# High-speed energy-efficient memristor confined in sub-5 nm space with elemental oxygen reservoir layer

Chenfei Li[1,4], Wencheng Niu[2,4], Da Wan [1]✉, Lin Tang [2], Zhengdao Xie[2], Kai Zhang[1], Yuan Liu [2], Qi Liu [3], Lei Liao [2]✉, Xuming Zou [2]✉ & Xingqiang Liu [2]✉

Random migration of oxygen vacancies ($V_O$) leads to unpredictable formation and rupture of conductive filaments (CFs) in oxide-based memristors. In this work, an atomically flat 4.5 nm hafnium oxide ($HfO_x$) switching layer and a 3.5 nm elemental oxygen reservoir (EOR) layer are confined between two-dimensional $HfS_2$ and $MoS_2$ layers, ensuring a homogeneous electric field distribution. The migration and redistribution of $V_O$ within the ultrathin $HfO_x$ switching layer enable the memristive behavior of the device. The EOR-based memristors achieve high set/reset transition speeds of 8 ns and 15 ns, respectively. The electroneutral EOR layer interacts with $V_O$ in the $HfO_x$ switching layer and, together with the $HfO_x$ tunnel layer above the $HfS_2$, forms a barrier to suppress the high-resistance state current. Reliable endurance up to $10^5$ cycles, and long retention up to $10^5$ s are simultaneously obtained. Finally, a high recognition accuracy of 97.0% is achieved, demonstrating potential for low-power neuromorphic computing applications.

Memristors are two-terminal devices featuring resistance realized through mobile species, typically oxygen vacancies ($V_O$), in an oxide-based switching layer. These species can be driven back and forth to create conductive filaments (CFs)[1–5]. Under an electric field, the drift of mobile $V_O$ causes the memristor to switch from a high-resistance state (HRS) to a low-resistance state (LRS). Thus, the resistive switching behavior of oxide-based memristors is governed by the growth and rupture of nanoscale CFs[6–9]. The LRS in such devices is determined by the formation of a percolation path between the electrodes, while data retention refers to the time required for these CFs to dissipate under ambient conditions[10–12]. However, irregular migration paths and the continuous accumulation of $V_O$ impact the uniformity of the conductive path during the set process, and increase variation in the rupture process during the reset process[13]. The expanded space for $V_O$ migration and redistribution during cycling gives rise to uncertainties and complications, resulting in eventual failure or irreversibly

degraded memristive switching[14]. Meanwhile, as the set/reset speed is limited by the electrical charging time and the random migration of ions, it is challenging to control the diffusion coefficients or paths of ions within the switching oxide thin films[8,15,16]. Although ultrathin switching layers offer an alternative route to minimize power consumption and increase operation speed, the morphology and scale of the CFs are significantly affected when the thickness is scaled down, making the control of CF growth/rupture more difficult. The presence of residual CFs generates a large HRS current, leading to additional static power consumption. Especially when the switching layer thickness is below 10 nm, surface roughness fluctuations impact the growth/rupture of CFs and increase the uncertainty of the switching process. Therefore, it is urgent to design ultrathin material stacks with a uniform structure and desirable interfaces to ensure a homogeneous electric field distribution across the switching layer, which can provide predictable migration paths[2,8,16]. Additionally, in forming-free oxide-

[1]School of Electronic Information, Wuhan University of Science and Technology, Wuhan, China. [2]State Key Laboratory for Chemo/Biosensing and Chemometrics, College of Semiconductors (College of Integrated Circuits), Hunan University, Changsha, China. [3]The Frontier Institute of Chip and System, Fudan University, Shanghai, China. [4]These authors contributed equally: Chenfei Li, Wencheng Niu. ✉e-mail: wanda@wust.edu.cn; liaolei@whu.edu.cn; zouxuming@hnu.edu.cn; liuxq@hnu.edu.cn

based memristors, previous works have primarily focused on modulating oxygen content through bilayer designs[11,17]. However, the uncontrolled presence of other unwanted defects leads to charge trap formation, which in turn results in high leakage currents and poor cycling stability[14,15,18,19]. Furthermore, a defective metal-insulator interfacial layer may induce defects to extend into the switching layer, creating parasitic series connections[17]. Consequently, it is difficult to maintain a high on-off ratio and low leakage current while reducing the thickness of the switching layer[20–23]. Notably, by confining the stochastic switching layer between two atomically flat interfaces, stochastic diffusion at the interfaces can be suppressed during operation under electrical fields[8,15]. Here, we demonstrate that an electroneutral elemental oxygen reservoir (EOR) layer can form an atomically flat interface with an ultrathin oxide layer. It acts as a tunneling barrier to block the stochastic migration of $V_O$ and suppress leakage current[15]. As a result, the integration of an EOR layer with the switching oxide layer enables highly stable memristor operations with ultralow power consumption by suppressing leakage current.

In this article, a 4.5 nm hafnium oxide ($HfO_x$) switching layer and a 3.5 nm electroneutral EOR layer are sandwiched between $HfS_2$ and $MoS_2$ flakes to form an ultrathin memristor structure. The ultrathin $HfO_x$ switching layer features a uniform structure and is derived from the $HfS_2$ flake through an ozone treatment, utilizing the atomically flat

surface of the $MoS_2$ flake as a substrate. This configuration enables ultrafast and low-power memristive operations. The memristor exhibits a high on-off ratio of $1.47 \times 10^9$ and an ultralow average HRS current of $1.52\,fA\mu m^{-2}$ along with ultra-short set/reset transition times (8 ns/15 ns). Furthermore, we demonstrated handwriting recognition using a memristor-based neural network model on the Modified National Institute of Standards and Technology (MNIST) dataset. By adjusting the pulse amplitude to update the weight parameters, we achieved a recognition accuracy of 97.0%. This work paves the way for the development of high-performance, energy-efficient computing and intelligent systems.

## Results and Discussion
### Fabrication of elemental oxygen reservoir-based transistor

The fabrication processes are depicted in Fig. 1a–c. In brief, a few-layered $MoS_2$ flake is mechanically exfoliated using Scotch tape and transferred onto an Cr/Au electrode on a $SiO_2$/ p+-Si substrate. Subsequently, a 2D $HfS_2$ flake with a thickness of ~12 nm is stacked onto the $MoS_2$ flake to assemble an $HfS_2$/$MoS_2$ van der Waals junction. Then, an ozone treatment is introduced to convert the $HfS_2$ flake into an $HfO_x$/$HfS_2$/$HfO_x$ functional stack. The oxidation process of $HfO_x$ can be precisely controlled by adjusting the oxidation time, and the evolution of the $HfS_2$ flake under ozone treatment is illustrated in Supplementary

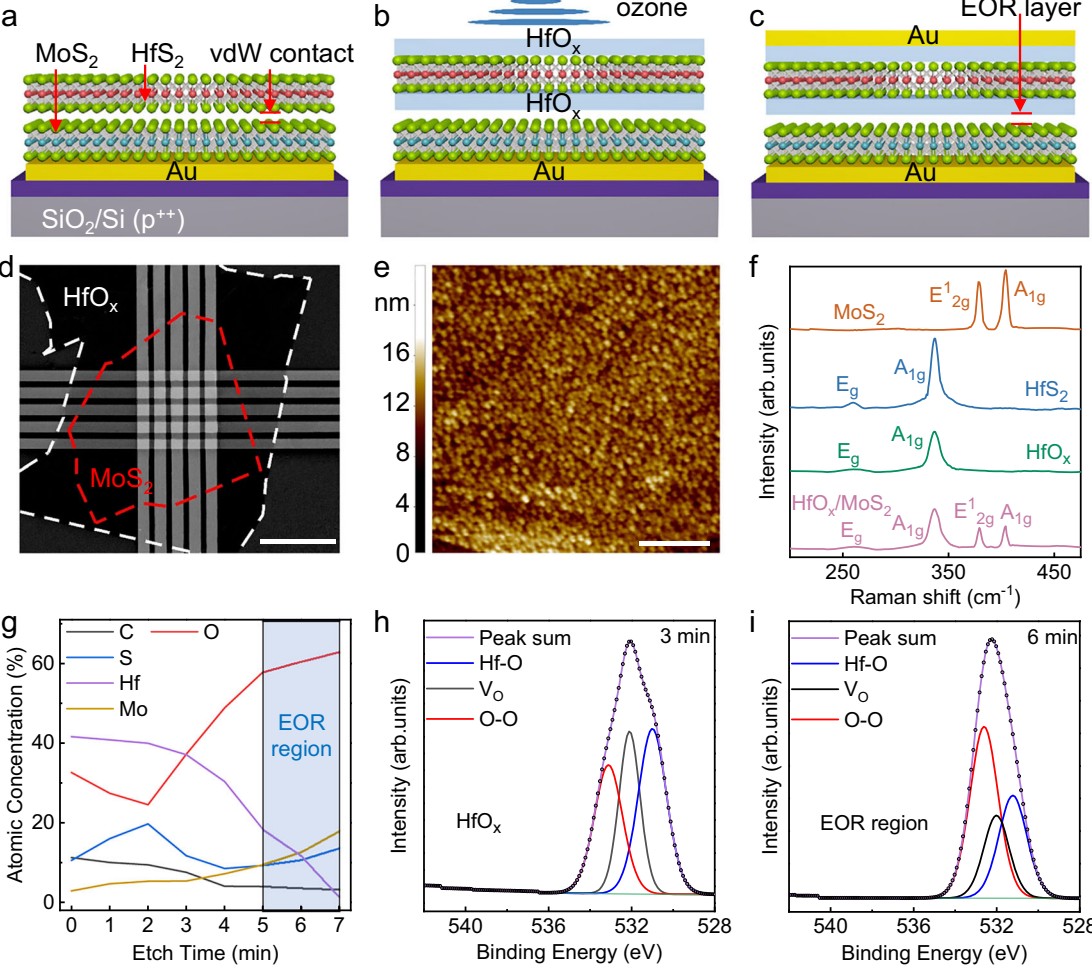

**Fig. 1 | Device structure and characterization. a** $MoS_2$ flake is transferred onto pre-fabricated Cr/Au (10/50 nm) electrode. **b** $HfS_2$ is stacked onto $MoS_2$ flake, and ozone treatment is introduced to transform $HfS_2$ into $HfO_x$/$HfS_2$/$HfO_x$ functional layer. **c** Top Au electrode (50 nm) is formed. **d** SEM image of the EOR-based memristor array. The scale bar is 20 μm. **e** AFM image of the $HfO_x$ surface. The scale bar is 5 μm. **f**, Raman spectra of different regions of the fabricated memristor. **g** Plots of elemental concentrations versus etching time. **h, i**, Depth-resolved O 1s XPS spectra of EOR-based memristor measured after (**h**) 3 min and (**i**) 6 min of etching.

Fig. 1. An atomic force microscopy (AFM) image indicates a smooth surface, and the thickness variation of the stack across different ozone treatment durations is presented in Supplementary Fig. 2. The increased height of the $HfO_x$/$HfS_2$/$HfO_x$ layer stems from the transition of the layered crystalline $HfS_2$ flake into an amorphous oxide layer. Finally, photolithography, thermal evaporation, and lift-off processes are carried out in sequence to form the top contact leads, as shown in Fig. 1c. An EOR layer is formed between the $HfO_x$ switching layer and the bottom $MoS_2$ flake, which originates from the van der Waals contact of the $MoS_2$/$HfS_2$ stack. Figure 1d presents the SEM image of the as-fabricated vertical Au/$MoS_2$/$HfO_x$/$HfS_2$/$HfO_x$/Au EOR-based memristor, in which the active area of each memristor is 4.2 μm × 4.2 μm, defined by the overlap region between the top and bottom electrodes. Since a mild ozone treatment is employed as a substitute for a relatively high-energy oxidation process, a mean roughness of 2.61 nm is obtained after oxidation for 1 h, as shown in Fig. 1e. Although ozone treatment involves reactive species and may alter the $MoS_2$ surface, $HfS_2$ flake fully overlaps the $MoS_2$ flake and the contact regions, suppressing unintentionally introduced defects and the influence of ultraviolet light illumination on $MoS_2$. Consequently, the obtained $HfO_x$/$MoS_2$ interface is atomically steep. The Raman spectra of different regions of the fabricated memristor are shown in Fig. 1f. The degraded $E_g$/$A_{1g}$ phonon modes confirm the transition to amorphous $HfO_x$, while the slightly shifted $E^1_{2g}$/$A_{1g}$ peaks of $MoS_2$ can be attributed to the doping effect induced by the EOR layer. We conducted a detailed depth-resolved X-ray photoelectron spectroscopy (XPS) characterization of the EOR-based memristors and analyzed the bonding state of oxygen in different layers, as shown in Fig. 1g. The content of high binding energy oxygen, corresponding to O-O bonds in the EOR-based memristors, is relatively high on the $HfO_x$ surface, which may be caused by surface adsorbed oxygen, as shown in Supplementary Fig. 3a. Since the XPS spectral signal is collected within a depth of ~5 nm, the Mo and S contents simultaneously increase after 5 minutes of etching, as shown in Fig. 1g, suggests that the XPS spectra obtained when the etching time reaches 6 minutes mainly originate from the EOR layer. The S concentration peaks at an etching time of 2 minutes, indicating that the spectrum obtained after 3 minutes of etching represents the $HfO_x$ switching layer. The intensity of $V_O$ rises in the $HfO_x$ switching layer, as shown in Fig. 1h. Notably, the O-O signal is dominant in the EOR region, while the $V_O$ and Hf-O bonds significantly decrease, as shown in Fig. 1i. Therefore, the EOR layer is mostly composed of O-O bonds. Moreover, O-O related species, such as $O_2^{2-}$, $O_2^-$, and $O_3$, possess low bond orders and are metastable. Thus, they are highly prone to gaining or losing electrons to achieve a more stable $O_2$ electronic configuration. Metallic hafnium is observed in the $HfO_x$ switching layer, as shown in Supplementary Fig. 3b. Typically, metallic hafnium promotes the formation of local $V_O$-related CFs, thereby reducing the forming voltage and even enabling forming-free operations. Furthermore, as shown in Supplementary Fig. 3c, the absence of Mo $3d$ features indicative of Mo oxidation in the XPS spectrum suggests that no Mo-O-related species were formed. Consequently, the EOR layer is mainly composed of electroneutral $O_2$.

## Electrical performance

To further characterize the switching capability of the vertical EOR-based memristor, electrical measurements are conducted using quasistatic d.c. sweeping and dynamic electrical pulses. The $MoS_2$/$HfS_2$ stack is fabricated in a nitrogen-filled glove box via mechanical exfoliation. Due to the low oxygen vacancy concentration in the unoxidized state, negligible memristive behaviors are observed, as shown in Supplementary Fig. 4. As the thickness of the $HfO_x$ switching layer is determined by the ozone treatment time, Fig. 2a presents the oxidation-time-dependent current-voltage (I-V) curves of the vertical EOR-based memristors, which clearly demonstrate stable bipolar resistive switching characteristics. A systematic reduction in the off-

state current density is observed with increased oxidation time. Accumulated oxygen tends to adsorb onto the $HfO_x$ surface and can recombine with $V_O$ in $HfO_x$ near the EOR layer. Consequently, the effective thickness of the $HfO_x$ switching layer may be thinner than the actual physical thickness. Combined with the metallic hafnium content and high $V_O$ concentration, the ultrathin $HfO_x$ utilizes the intrinsic electronegativity gradient to drive oxygen redistribution, allowing the memristor to operate in a forming-free mode. The slightly inconsistent set voltages may be attributed to thickness variations in the $HfO_x$ flakes during the oxidation process (Fig. 2b). Since the process of oxidizing $HfS_2$ into $HfO_x$ is not self-limiting, increasing the oxidation time (ozone treatment time) increases the $HfO_x$ thickness. However, this leads to a continuous increase in the HRS current after 40 min, as shown in Fig. 2b. This is mainly because as the thickness increases, the number of defects also increases, significantly raising the probability of tunneling through the EOR layer. Moreover, the values of $V_{SET}$ and $V_{RESET}$ also constantly increase when the ozone treatment exceeds 40 min, which is mainly because the increased thickness requires a higher operating voltage. Therefore, reducing the thickness of the switching layer can decrease the power consumption of the device by lowering the operating voltage. The film thickness reaches saturation after 80 min of ozone treatment. However, prolonged exposure leads to the degradation of the surface morphology, resulting in a marginal increase in the SET and RESET operating voltages. The LRS current is primarily determined by the confined CFs formed by $V_O$ within $HfO_x$ regions, so the LRS current remains stable. Typical set/reset cycles of 100 vertical EOR-based memristors from four 5 × 5 memristor arrays are presented in Fig. 2c. The corresponding I-V curves with absolute current values are provided in Supplementary Fig. 5. Since the electroneutral EOR layer functions as a barrier for leakage current, the potential defect-assisted current in the HRS state is suppressed. According to the statistical cumulative probability distribution of current at HRS and LRS in Fig. 2d, an ultra-low average HRS current density of 1.52 fA μm$^{-2}$ and a high average on-off ratio of $1.47 \times 10^9$ at a read voltage ($V_{Read}$) of 0.1 V are simultaneously achieved (Supplementary Fig. 6a). Statistical analysis reveals that the set and reset voltages ($V_{SET}$ and $V_{RESET}$) follow Gaussian distributions centered at 1.2 V and −0.8 V, respectively (Supplementary Fig. 6b).

The electroneutral EOR layer also contributes to the robust retention performance of the fabricated memristor, as shown in Fig. 2e. At $V_{READ} = 0.1$ V, stable HRS/LRS currents are maintained for $10^5$ s at room temperature without degradation. Testing stability at high temperature is necessary to illustrate the performance of the EOR-based memristor. As shown in Supplementary Fig. 7, reliable and consistent switching characteristics over $10^5$ s are obtained at 85 °C and 125 °C, respectively. Typical endurance performance of the fabricated memristors at room temperature is shown in Fig. 2f. Under continuous pulses of ±1.5 V/200 μs, the LRS and HRS exhibited remarkably stable switching over $10^5$ cycles, with both HRS and LRS currents remaining constant and showing minimal fluctuations. Figure 2g presents the detailed response under consecutive pulse excitation. Although a 100 ns pulse is insufficient to drive the LRS current of the EOR memristor to a large saturation value, such as approaching the compliance current, an on-off ratio of ~$10^4$ is still obtained. This indicates that such a short pulse is sufficient for the growth of CFs and that the device can operate normally under short pulse conditions without performance degradation after $10^5$ cycles (Supplementary Fig. 8). To investigate the kinetics of forming-free switching, we performed pulse-width-dependent experiments, varying the pulse width from 10 ns to 100 ns. As shown in Supplementary Fig. 9, the LRS current increases with the increase of pulse width. The switching time of the device is evaluated by V-t and I-t synchronous curves measured in pulse mode. Figure 2h shows the oscilloscope waveforms of the applied voltage pulses (±1.5 V/100 ns) and the measured current during set/reset operations. A constant bias of 0.1 V (read voltage) is used

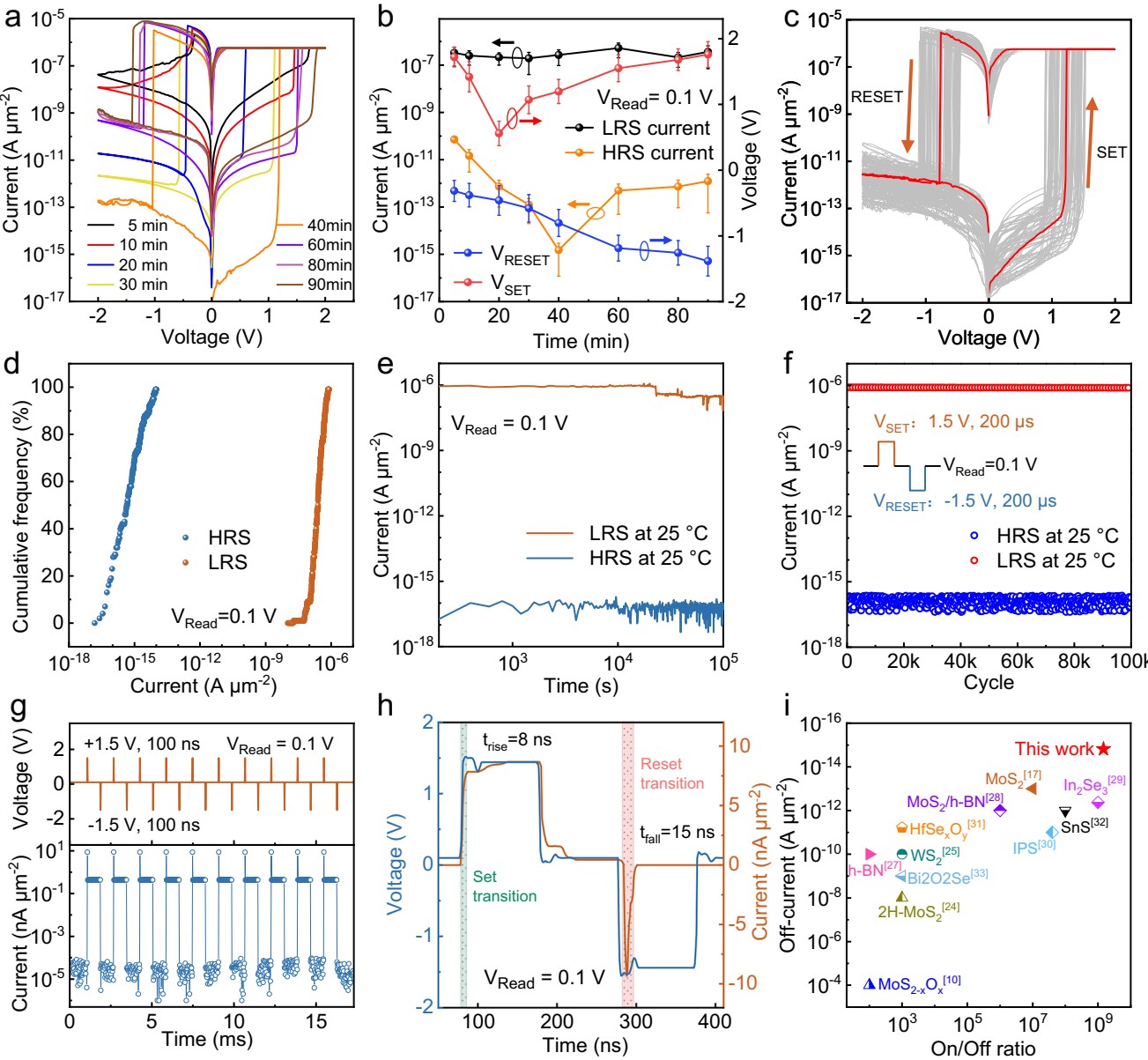

**Fig. 2 | Electrical performance of EOR-based memristor arrays. a** Influence of ozone treatment time on the electrical performance of the fabricated EOR-based memristors. **b** Plots of set/reset voltage and LRS/HRS currents versus oxidation time of the EOR-based memristors. The error bars represent the standard deviation calculated from 100 independent devices. **c** Typical bipolar I-V curves of 100 EOR-based memristors from four 5 × 5 arrays. **d** Cumulative probability distribution of the resistance in the HRS and LRS extracted at $V_{READ} = 0.1$ V. **e** Retention performance of the fabricated memristor with an ozone treatment time of 40 min. **f** Corresponding endurance characteristics of the fabricated memristors, where the cycling pulse for set/reset is ±1.5 V with a width of 200 μs. **g** Variation of current values after set (+1.5 V, 100 ns) and reset (−1.5 V, 100 ns) pulse operations. **h** Set/reset voltage pulses (±1.5 V, 100 ns) and transient current response. **i** Comparison of the on-off ratio and off-current of ultrathin EOR-based memristors.

to measure the current. The induced current takes 8 ns to increase from 10% to 90% of its maximum value, which can be characterized as the set transition time of the EOR-based memristor. The reset pulse gradually pulls down the current as filament dissolution proceeds, and a reset transition time of 15 ns is obtained. The performance parameters of the fabricated memristor demonstrate a high on-off ratio and an ultralow off-current compared with previously reported ultrathin memristors, as shown in Fig. 2i. This work presents a feasible method to fabricate atomically thin an $HfO_x$ switching layer without invasive sputtering or atomic layer deposition. Moreover, in the fabricated memristor, no forming process is required to initiate resistive switching and create CFs, making it advantageous in terms of forming-free operation compared to previously reported ultrathin oxide memristors, as shown in Supplementary Table 1[10,17,24–33].

## Operation mechanism

To investigate the switching mechanism and ultralow HRS current of the fabricated EOR-based memristor arrays, electron energy loss spectra (EELS) line scans are carried out across the EOR-based memristor, enabling direct visualization of the oxygen distribution in the vertical stack. This visualization elucidates the structural and compositional variations of the device. This distinct O K edge at 532 eV reveals the partial density of states of elemental oxygen, and the shaded region represents the O K partial density of states summed from different layers, indicating the presence of an EOR layer between the $MoS_2$ and the switching $HfO_x$ layer, as shown in Fig. 3a. Energy loss near edge structure (ELNES) spectra are shown in Supplementary Fig. 10a. The O K-edge of $HfO_x$ is dominated by two peaks labeled p1 and p2, centered at 534.5 eV and 538.1 eV, respectively. Typically, p1-p2 peak

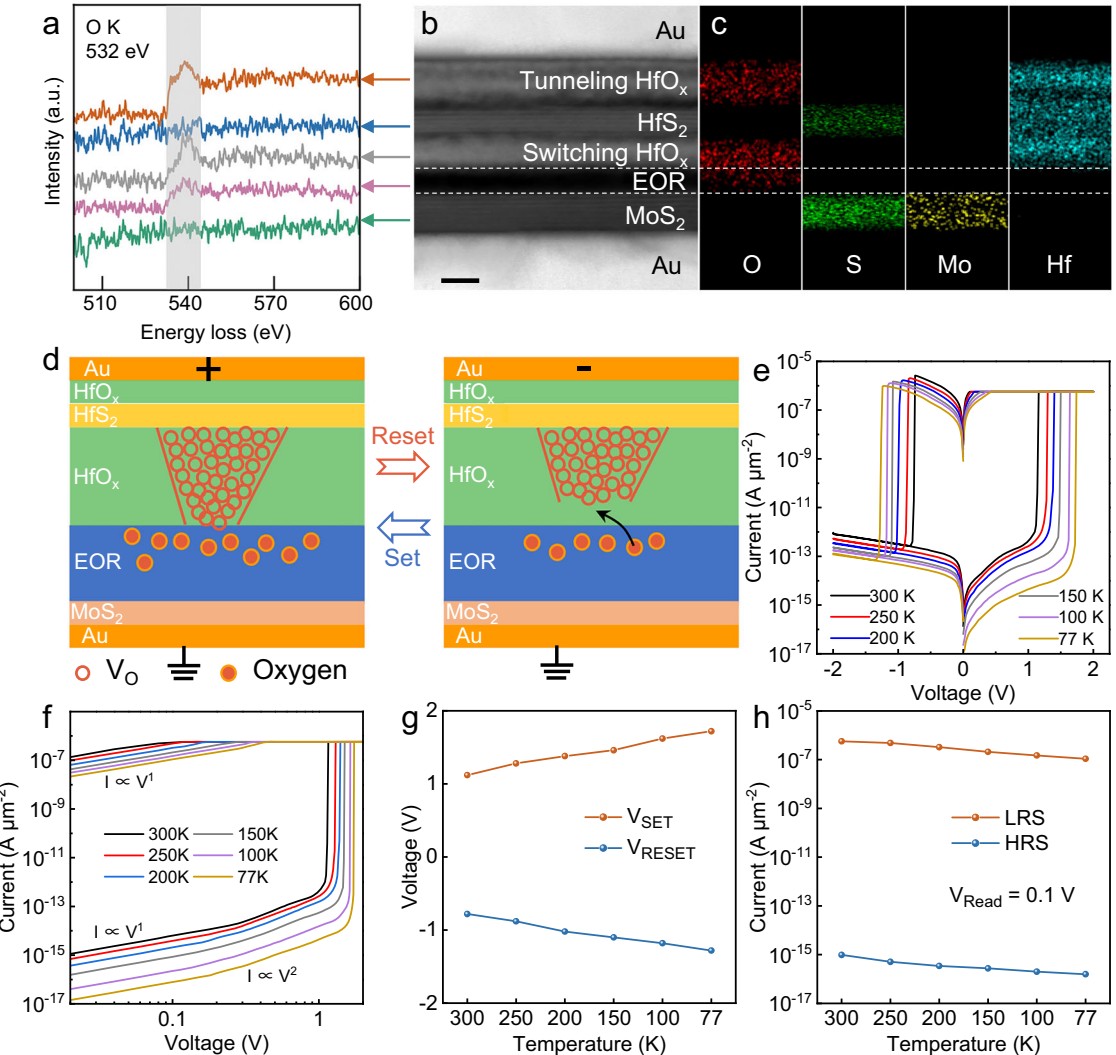

**Fig. 3 | Resistive switching mechanism of EOR-based memristors. a** EELS spectra of O K core-level edges extracted from different layers. **b, c** Cross-sectional TEM image and corresponding elemental distribution. The scale bar is 5 nm. **d, e** Schematic illustration of the operation mechanism of the EOR-based memristor in the HRS and LRS. **e** The I-V curve of the EOR-based memristor measured at different temperatures. **f** The set I-V curve of the EOR-based memristor in double logarithmic coordinates. **g, h** Temperature-dependent electrical performance of the switching voltages and the LRS-HRS currents. $V_{Read}$ is set to 0.1 V to reflect the trend of HRS current with temperature.

splitting is absent in transition metal oxide films with excess oxygen. The O K-edge of the EOR layer exhibits an obvious single peak at 538.3 eV. Therefore, these results confirm the oxygen-rich nature of the $HfO_x$ and EOR layer in the stack[34–37]. Transmission electron microscopy (TEM) and energy dispersive spectroscopy (EDS) mapping are shown in Fig. 3b, c, and the thicknesses of the switching $HfO_x$ layer and the EOR layer are 4.5 nm and 3.5 nm, respectively. The EOR layer originates from the van der Waals contact of the $HfS_2$/$MoS_2$ interface, where the oxygen-rich condition leads to oxygen atoms to accumulate at the interface, allowing the $HfS_2$ flake to be partly oxidized from both sides. Since the atomically flat surface of $MoS_2$ provides an flat platform for the growth of the $HfO_x$ switching layer, the vertical architecture exhibits atomic flatness at the interface (Supplementary Fig. 10b). This minimizes reliability issues caused by thickness fluctuations, especially in the case of sub-10 nm thickness. Moreover, this interface limits unwanted charge trapping, which helps stabilize the local electric field distribution and improves the uniformity of the switching process. Therefore, the ultra-flat $HfO_x$ switching and EOR layers ensure a homogeneous electric field distribution, enabling stable resistive switching operations.

The bipolar resistive switching characteristics of the EOR-based memristor originate from the synergistic interplay of $V_O$ migration within the $HfO_x$ switching layer located between the $HfS_2$ and EOR layers, as schematically illustrated in Fig. 3d. Briefly, a positive bias (set) creates an electric field that induces a simultaneous soft breakdown via field-driven $V_O$ percolation within the vacancy-rich $HfO_x$ switching layer. As a result, CFs are formed, and the memristor is set to the LRS. The opposite effect occurs when applying a negative voltage (reset). Oxygen ions at the $HfO_x$/EOR interface recombine with $V_O$, rupturing the CFs, and the memristor is reset to the HRS with negligible residual filaments. The $HfO_x$ tunneling layer above the $HfS_2$ acts as a series tunneling barrier to suppress crosstalk, helping to mitigate excessive current flow. Sneak-path analysis is presented in Supplementary Fig. 11, where the voltage pulsing scheme involves applying a designated voltage to the selected cell and smaller voltages to the half-selected cells due to the standby voltage scheme. The selected cell experiences conductance changes under external pulses, while other regions of the crossbar arrays remain largely unaffected. Moreover, intended digital patterns are successfully stored and read out, as shown in Supplementary Fig. 12, demonstrating that the readout of neighboring cells is

not disturbed by sneak currents. This result confirms that crosstalk in the proposed memristor array is effectively suppressed and that each cell can be individually addressed and read without unintended interference from adjacent devices. Since the $MoS_2$ and $HfS_2$ layers present high energy barriers for $V_O$ migration, and the EOR layer prevents oxygen drift and diffusion, relatively thin CFs are formed in the $HfO_x$ layer. This helps reduce overcurrent stress, facilitating low-current operation. The $HfO_x$ switching layer and EOR layer are confined within a nanoscale space, and stochastic diffusion at the interface is effectively suppressed.

The I-V curves in Fig. 3e demonstrate bipolar nonvolatile switching from 77 K to 300 K, revealing that temperature significantly influences the resistive switching behavior. As shown in Fig. 3f, the slope of the fitting lines in the low voltage region of the LRS demonstrates linear $I \propto V^1$ behavior, which is consistent with Ohmic conduction through percolating CFs formed under electric field-driven $V_O$ migration. Defect-related traps in the switching oxide layer contribute to a slightly higher slope (greater than 1). The slope of the fitting lines increases to ~2 at higher voltage, which is correlated with an increase in trap concentration within the band gap. Since Child's Law ($I \propto V^2$) applies to the space charge limited conduction (SCLC) mechanism, SCLC and Ohmic conduction coexist to govern the HRS conduction mechanism, whereas Ohmic conduction dominates the LRS conduction mechanism. Specifically, conduction in the HRS transitions from Ohmic behavior ($I \propto V$) at low biases to SCLC behavior at higher biases, following Child's law. The transition to SCLC signifies the dominance of injected carriers over the intrinsic defect-induced charge density. Figure 3g plots the set and reset voltages versus temperature. Decreased migration efficiency hinders the formation and dissolution of CFs at low temperatures, resulting in higher switching voltages for the memristor. The temperature-dependent current plots for HRS and LRS at $V_{Read} = 0.1$ V are shown in Fig. 3h. Based on the Arrhenius plot with linear fitting shown in Supplementary Fig. 13, the barrier height is extracted to be 15.4 meV. Owing to the uniform and ultrathin $HfO_x$ switching layer confined within a sub-10 nm space, a homogeneous electric field distribution is realized across the device. The EOR layer and $HfS_2$ flake act as diffusion barriers to inhibit the random diffusion of $V_O$, leading to robust endurance and retention performance of the EOR-based memristors.

## Artificial neural networks

The development of biologically plausible artificial neural networks (ANNs) necessitates the precise emulation of synaptic plasticity through memristive architectures, as shown in Fig. 4a. A memristor crossbar array is usually employed to perform vector-matrix multiplication for pattern classification. The detailed multilayer perceptron (MLP) structure is shown in Fig. 4b, comprising 784 input neurons, 300 hidden neurons, and 10 output neurons, all interconnected by device-simulated synapses. Digital images of different handwritten digits are imported into the first layer of neurons and are processed by the MLP algorithm. This synaptic analog enables the construction of a network with backpropagation-tunable weights, as shown in Fig. 4c, which eliminates von Neumann bottlenecks through analog vector-matrix multiplication within the memristor array. Synaptic validation commenced with paired-pulse experiments. Figure 4d, e illustrates the paired-pulse facilitation (PPF) and paired-pulse depression (PPD) characteristics, respectively. The PPF ratio decreases with increasing time intervals between pulses, following a double exponential decay function, while the PPD ratio increases with the time interval, also following a double exponential function. These bidirectional time-dependent responses mirror biological short-term plasticity governed by presynaptic calcium dynamics. The memristor exhibits frequency-dependent current modulation under varying pulse train frequencies ($10^2$–$10^4$ Hz), confirming its adaptability to various stimulation patterns (Fig. 4f). This adaptive signal transduction capability underscores the

device's potential for implementation in complex neural network applications. This memristor exhibits long-term potentiation (LTP) under high-frequency activity and long-term depression (LTD) under low-frequency activity, as shown in Fig. 4g, h. The simulation results, as shown in Fig. 4i, indicate that the trained neural network achieves a recognition accuracy of 97.0% for the device using non-identical pulses and 91.5% for the device using identical pulses, approaching the ideal simulation value of 97.4% based on the MNIST dataset. Therefore, the EOR-based memristor array shows great potential for neural network systems.

By taking advantage of the atomically flat interface of 2D materials, a uniform and ultrathin $HfO_x$ switching layer is fabricated between $HfS_2$ and $MoS_2$. Consequently, the migration of oxygen vacancies is confined within a sub-10 nm space, ensuring a homogeneous electric field distribution. The EOR layer acts as a diffusion barrier and facilitates the rupture of CFs in the HRS, enabling ultrafast operation and low power consumption. Furthermore, the EOR-based memristors exhibit desirable retention and endurance performance. Moreover, forming-free operation is achieved due to the ultrathin nature, high $V_O$ concentration, as well as the presence of metallic hafnium in the switching layer. Simulation results demonstrate that the EOR-based memristor array obtains a high recognition accuracy, indicating its potential for low-power memory and neuromorphic computing applications.

## Methods
### Device Fabrication
The EOR-based memristor arrays were fabricated on a $SiO_2/p^+$-Si substrate. The bottom Au electrodes (50 nm) with a Cr adhesion layer (10 nm) were prefabricated using standard mask photolithography, thermal evaporation, and lift-off processes. Multilayer $MoS_2$ flakes were mechanically exfoliated from a bulk crystal (Shanghai Onway Technology Co., Ltd) using Scotch tape and then transferred onto the prefabricated Au electrodes. Subsequently, $HfS_2$ flakes with a thickness of ~12 nm, also mechanically exfoliated, were stacked onto the $MoS_2$ flakes to form an $HfS_2/MoS_2$ van der Waals heterostructure. The functional $HfO_x$ switching layer and the elemental oxygen reservoir (EOR) layer were created by subjecting the $HfS_2/MoS_2$ assembly to a mild ozone treatment. The thickness of the resulting $HfO_x$ layer was precisely controlled by adjusting the duration of the ozone exposure. The ozone was produced by ultraviolet light in a reaction chamber (Model BZS250GF-TS) pre-charged with oxygen gas under ambient pressure conditions. A 300 W ultraviolet lamp was employed as the excitation source, delivering an ultraviolet intensity of 50 mW $cm^{-2}$. Meanwhile, thermal irradiation regulated the substrate temperature at a mild 310 K. Methyl methacrylate/poly methyl methacrylate (MMA/PMMA) bilayer was employed as the photoresist material, and EBL was utilized to define the geometric pattern of top electrodes. Finally, the top Au electrodes (50 nm) were defined by a final round of thermal evaporation to complete the vertical EOR-based memristor devices.

### Material and device characterizations
The morphology of the device was imaged with an optical microscope (Olympus BX51M) and a scanning electron microscope (JEOL IT300). The surface roughness of the $HfO_x$ layer after the ozone treatment was evaluated with an atomic force microscope (Park NX20). Raman spectroscopy (Witec alpha300R) was performed to confirm the structural changes. X-ray photoelectron spectroscopy (XPS) measurements were carried out using a PHI 5000 VersaProbe III system (ULVAC-PHI, Inc.) with a monochromatic Al Ka source (1486.6 eV). High-resolution spectra of Hf $4f$, Mo $3d$, and O $1s$ were collected, and all binding energies were calibrated using the C $1s$ peak at 284.8 eV. A multibeam scanning electron microscopy–focused ion beam system (JIB-4501, JEOL) was employed for cross-sectional sample preparation, with operation performed using a 30 keV $Ga^+$ ion beam. Specifically, a

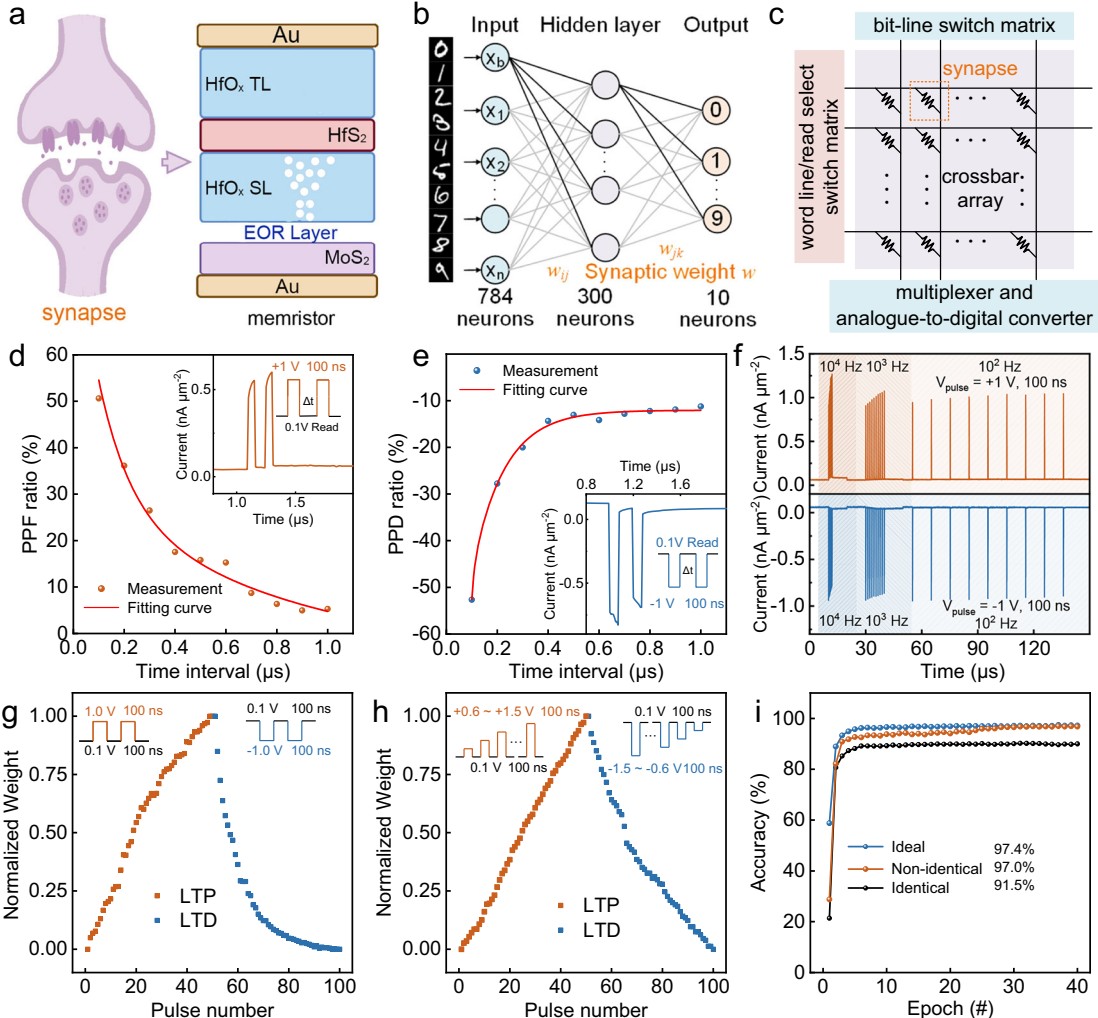

**Fig. 4 | Simulation implementation of image recognition based on EOR-based memristors. a** Schematic of a biological synapse and the corresponding artificial synapse. **b** Schematic diagram of the three-layer ANN structure. **c** Conceptual PIM architecture of the artificial synapse array. **d, e** PPF and PPD characteristics of the EOR-based memristor, respectively, measured at $V_{Read}$ = 0.1 V. The inset shows two pairs of response currents read at $V_{Read}$ = 0.1 V. **f** Current response under pulse trains of varying frequencies. **g** LTP and LTD characteristics obtained with consecutive identical pulses (±1 V, 100 ns). **h** Adjusted LTP and LTD characteristics using consecutive non-identical set and reset pulses. **i** Evolution of MNIST test image recognition accuracy versus training epochs for an ideal device and the EOR-based memristor with identical and non-identical pulses.

1 μm-thick protective layer of platinum or carbon was deposited in situ prior to ion-beam irradiation. The target region was then milled using a 30 kV accelerating voltage with a gallium ion beam current of 2.8-47 nA to obtain a thick lamella. The lamella was subsequently lifted out from the original substrate, mounted onto a STEM half-grid using a micromanipulator, and further thinned to <100 nm using a 24–790 pA gallium ion beam to prepare an electron-transparent specimen for STEM observation. All procedures were conducted in a vacuum chamber maintained at ~$10^{-4}$–$10^{-5}$ Torr. For a detailed cross-sectional analysis, a transmission electron microscopy (JEOL JEM-2100F TEM/scanning TEM instrument) operated at 200 kV, equipped with EDS (Oxford INCA) mapping and EELS (Gatan Enfina), was used to visualize the layer thickness, elemental distribution, and the presence of the oxygen-rich EOR layer.

## Device electrical measurements
All electrical measurements were performed under ambient conditions except the temperature measurement. The current-voltage (I-V) characteristics, endurance, and retention tests were performed using a Lake Shore TTPX Probe Station with Keysight B1500A and Agilent B2912A semiconductor parameter analyzers. Quasistatic direct current sweeps and dynamic electrical pulses generated by a Keithley 4200 SCS were applied to investigate the memristive properties. The switching speed of the device was evaluated using a pulse generator (Keithley 3402) and an oscilloscope (GDS-1102B). The retention performance was measured at room temperature with a read voltage of 0.1 V. Temperature-dependent measurements were performed in a vacuum probe station under a pressure of ~0.1 Pa at temperatures ranging from 77 K to 300 K.

## Neural Network Simulation
A multilayer perceptron (MLP) model, consisting of 784 input neurons, 300 hidden neurons, and 10 output neurons, was implemented in Python. The experimentally obtained long-term potentiation (LTP) and long-term depression (LTD) characteristics of the memristor were used to update the synaptic weights in the network. The simulation was conducted using the Modified National Institute of Standards and Technology (MNIST) dataset, and the recognition accuracy of the network was calculated over 40 training epochs to evaluate the learning capability of the system.

## Data availability

The relevant data generated in this study are provided in the Source Data file. Relevant data supporting the key findings of this study are available within the article and the Supplementary Information file. All raw data generated during the current study are available from the corresponding authors upon request. Source data are provided with this paper.

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

## Acknowledgements

This work was supported by the National Key Research and Development Program of Ministry of Science and Technology (No. 2022YFA1402504), the National Natural Science Foundation of China (Grant Nos. 92464303, U24A20302, U22A2074, 62305110, 62374054, 62474061, and 62274060), the Foundation for Innovative Research Groups (No. 62321003), the Natural Science Foundation of Hunan Province (2023JJ40164, 2025JK2017).

## Author contributions

C.L. and W.N. contributed equally to this work. L.L. and X.L. designed the research. C.L. prepared and characterized the materials and related devices. W.N., L.T., Z.X. and K.Z. participated in the experiments. X.Z., D.W., Q.L. and Y.L. discussed the experiments and provided suggestions. C.L., W.N. and X.L. co-wrote the manuscript with inputs from all the authors. All authors discussed the results and commented on the manuscript.

## Competing interests

The authors declare no competing interests.
