## [Transparent Peer Review file · Nature Communications]

High-speed energy-efficient memristor confined in sub-5 nm space with elemental oxygen reservoir layer

Corresponding Author: Professor Lei Liao

Version 0:

Reviewer comments:

Reviewer #1

(Remarks to the Author)

In this work, a 4.5 nm HfO_x switching layer is confined between an HfS₂ layer and a 3.5 nm elemental oxygen reservoir (EOR) layer. The device achieves ultrahigh Set/Reset switching speeds of 8 ns and 15 ns, respectively, with a normalized switching energy of 51 aJ. The HfO_x tunneling layer above the HfS₂ layer serves as a leakage barrier, enabling an ultralow leakage current of approximately 0.05 fA/μm², a large on/off ratio exceeding 10¹⁰, and long retention endurance up to 10⁴ seconds. Furthermore, when modulated by pulse amplitude, the memristor demonstrates a high recognition accuracy of 97.0% in a convolutional neural network. Overall, this manuscript is well-organized and innovative. Considering that the overall quality, I recommend publication after a minor revision. Here are some suggestions for improvements:

- (1) The author can provide the electrical performance of the device when it is not oxidized? In Figure 2, the influence of ozone treatment time on the electrical performance of the fabricated EOR-based memristors is provided. What would the performance of devices without oxidation be like?
- (2) What role does the following MoS₂ material play? To prevent the oxidation of MoS₂, the author deliberately selected a larger HfS₂ material. Why is HfS₂ placed on top of MoS₂ before oxidation? Oxidizing HfS₂ first and then performing van der Waals assembly won't result in a flatter interface?
- (3) As shown in Figure 2, the performance of the device is good. How about the reproducibility? What is the difference between the EOR layer and the hafnium oxide layer? How can the EOR layer affect the performance of the fabricated memristor?
- (4) Figure 4 shows the comparison of the recognition accuracy of MNIST test images. Why are the starting points of the accuracy for the three comparisons different? How does the EOR-based memristor maintain desirable linearity?
- (5) It is not clear that how the ozone treatment is not affecting to MoS₂? How the authors control the treatment condition considering each sample has different HfS₂ thickness?
- (6) In particular, there is evidence for a clean vdW interface and EOR layer, which are novel insight relevant to a broad audience and to the best of my knowledge the first demonstration of the usage EOR layer to improve the memristor performance, details about the fabrication processes and the TEM measurements should be added in method section.
- (7) What are the parameters of the ozone treatment (i.e. chamber pressure, O₂ partial pressure, UV light intensity, ...)? How can it be ensured that the topmost MoS₂ layers are not oxidized, forming an MoO₃ interlayer? Free monoatomic oxygen is a radical which readily bonds to any nearby molecule. Thus, there are certainly no oxygen atoms at the HfO₂ surface, these are either O₂ molecules or ozone (O₃) molecules which are also highly reactive and thus likely bond to the HfO₂ surface. Thus, the enhanced concentration corresponds to molecules of either O₂, O₃ or an off-stoichiometric phase of hafnium oxide with an increased O concentration at the amorphous HfO₂ surface. Please modify the text in the main manuscript and the supplementary information accordingly.

Reviewer #2

(Remarks to the Author)

Reviewer Report

The authors report a van der Waals (vdW) heterostructure memristor that integrates an ultrathin HfO_x switching layer and an "electroneutral elemental oxygen reservoir (EOR)" between MoS₂ and HfS₂. The device exhibits forming-free operation, an ultralow leakage current density of approximately 0.05 fA μm⁻², an on/off ratio exceeding 10¹⁰, and nanosecond-level

switching speed (8 ns/15 ns) with a remarkably low switching energy of about 51 aJ. Such performance metrics are undoubtedly impressive and place the device among the best in the ultrathin or 2D-material-based memristor category. The manuscript is well-written, the experimental data are clean, and the proposed architecture is conceptually interesting. However, several central claims—particularly regarding the physical nature of the EOR layer and the mechanism of forming-free switching—are speculative and insufficiently supported by quantitative evidence. These gaps prevent the manuscript from reaching the level of mechanistic rigor expected in Nature Communications.

The main strength lies in its excellent device performance. The authors convincingly demonstrate that the vdW-derived EOR configuration can achieve extremely low leakage and high on/off ratio simultaneously, which has been a long-standing challenge in sub-5 nm oxide devices. The forming-free, ultrafast switching behavior and the low switching energy further highlight the potential of the approach for future low-power memory and neuromorphic computing. The device fabrication based on mild ozone oxidation of HfS₂ into HfO_x is elegant and avoids plasma damage or interface disorder typical of ALD or sputtered films. TEM and EELS analyses provide a visually persuasive structural picture of the vertical stack, and temperature-dependent I–V measurements help delineate conduction regimes. Overall, the data presentation is of high quality and suggests that the authors have achieved careful experimental control.

Despite these, reviewer suggests that the mechanistic interpretation requires much stronger evidence. The manuscript repeatedly refers to the presence of an “electroneutral elemental oxygen reservoir” at the MoS₂/HfO_x interface, which is said to act as a tunneling barrier and diffusion regulator. However, the experimental basis for calling this layer “elemental” or “electroneutral” is insufficient. The EELS O–K edge data only reveal the presence of oxygen at the interface but do not identify its bonding state, valence, or neutrality. Without complementary depth-resolved XPS or ELNES analysis, it is impossible to confirm whether the layer consists of elemental oxygen, an oxygen-rich HfO_v, or MoO_v phase. This uncertainty is critical because the novelty and physical mechanism of the work hinge on the assumption that this layer is chemically distinct from conventional oxide reservoirs. At present, this interpretation remains speculative rather than conclusive.

Similarly, the proposed “dynamic equilibrium between oxygen and vacancies” as the origin of forming-free switching is presented as a central mechanistic claim but is not directly proven. The data show forming-free I–V characteristics and repeatability, but no kinetic or temperature-dependent experiments are performed to extract activation energies or diffusion coefficients that would substantiate such an equilibrium model. Without such analysis, the claim of a self-equilibrating oxygen-vacancy process remains qualitative and speculative. Quantitative studies of pulse-width, temperature, or interval dependence are needed to establish whether the switching indeed arises from a balanced oxygen exchange rather than conventional filamentary behavior limited by thickness or compliance.

Another point of concern is the assertion that the EOR suppresses crosstalk in array configurations. The manuscript presents no actual crossbar array data, sneak-path analysis, or half-select stress experiments. The claim is extrapolated from single-device leakage characteristics, which is not sufficient to demonstrate system-level selectivity or array compatibility. Similarly, while the manuscript attributes the ultralow leakage to the tunneling-barrier function of the EOR, temperature-dependent conduction analysis is limited to qualitative SCLC and Ohmic fitting. Quantitative extraction of barrier heights or Schottky/Poole–Frenkel/Fowler–Nordheim parameters would provide much stronger mechanistic grounding. Without these analyses, alternative explanations such as reduced trap density, series resistance in HfS₂, or interface depletion cannot be excluded.

The reliability data are also limited relative to the claims. The authors describe “robust retention and endurance,” yet only 10³ cycles and retention up to 10⁵ s (~28 h) at room temperature are presented. These values, though reasonable for a proof-of-concept, do not justify the use of “robust” or “stable” at the level expected for nonvolatile memory publications in Nature Communications. Measurements of endurance up to 10⁵–10⁶ cycles, retention at elevated temperatures (85–125 °C), and read-disturb or bias-stress testing are necessary to validate the long-term stability claims.

The comparison with the ALD-HfO_x control device is helpful but not conclusive, as the two structures differ in fabrication method, oxidation time, and possibly thickness or interface roughness. Therefore, attributing the performance improvement solely to the EOR effect is premature. A head-to-head comparison with strictly identical geometries and electrode materials would be required to isolate the role of the EOR. Furthermore, the neuromorphic functionality—highlighted by a 97% MNIST recognition accuracy—is purely simulation-based. While the LTP/LTD data are experimentally measured, the neural network performance results from an offline simulation that does not include device-to-device variation, cycle-to-cycle drift, or retention degradation. Without hardware-in-the-loop or on-chip learning experiments, the neuromorphic claims should be presented more cautiously as “potential applications” rather than demonstrations.

In comparison to recent literature, the manuscript distinguishes itself by proposing a vdW-induced, neutral-oxygen interlayer rather than the conventional trilayer HfO_x/HfO₂/HfO_x or HfO₂/ZrO_{2-x}/HfO₂ stacks used as oxygen-vacancy reservoirs. Chemically, this approach could represent a new direction for interfacial control in 2D memristors. However, recent trilayer studies already demonstrate CMOS-compatible fabrication, array-level validation, and selector-free suppression of sneak paths—advantages that this work does not yet match. The novelty of a “neutral oxygen” reservoir is intriguing but must be proven chemically to stand as a clear advance over these prior works.

In conclusion, this manuscript presents a technically strong and potentially impactful concept with outstanding device performance. However, several mechanistic and interpretive claims are currently speculative or insufficiently evidenced. The work would benefit greatly from deeper chemical analysis of the EOR layer, quantitative conduction modeling, comprehensive reliability testing, and hardware-level neuromorphic validation. If these issues are addressed, the study could reach the broad and rigorous standard of Nature Communications. At its current stage, I recommend major revision.

The authors are encouraged to provide quantitative and chemical evidence to solidify their mechanistic interpretation and to moderate claims that are presently speculative.

Reviewer #3

(Remarks to the Author)

This manuscript investigates HfS₂-based memristive devices. However, the work does not demonstrate substantial advances in either device performance or mechanistic understanding. Although the paper claims a sub-5-nm switching region, this refers merely to the film thickness rather than the actual device footprint. I initially expected lateral device scaling, but the reported vertical dimension does not provide meaningful novelty.

The manuscript further claims “high-speed” switching, including the 8 ns / 15 ns switching times emphasized in the title and abstract. In reality, the devices are driven by 100 ns pulses, with the current merely reaching a steady level at 8 ns / 15 ns. This does not constitute stable switching. In the community, a switching speed is typically accepted only if the device can reliably toggle under pulses of that duration, and the subsequent voltage/current holding sequence confirms stable filament formation or rupture. The same issue applies to the purported “ultralow energy” operation, where only the portion of the current transient up to stabilization is considered—this is not an appropriate metric. The MNIST recognition demonstration based on simulations also does not introduce meaningful novelty.

Additional concerns:

1. Device area: The manuscript does not specify the device area, which is critical for evaluating performance. Only current density is reported, without the absolute current values.
2. Endurance testing: Endurance is typically evaluated using ~200 μ s pulses. What would the endurance look like under 8 ns / 15 ns pulses? The reported endurance of ~1,000 cycles is insufficient for practical applications; can it be improved?
3. Retention testing: Retention should not be evaluated only at room temperature. Measurements at 85 °C—or preferably 125 °C—are standard practice.
4. Memory window: The experimental results indicate a memory window of approximately 3–5 orders of magnitude, with noticeable variation across tests. This is far from the 10-order-of-magnitude window claimed in the manuscript.
5. Inconsistencies in resistance values: The reported high-/low-resistance states vary significantly across figures. In Fig. 2b they are 10⁻¹² / 10⁻¹⁶ A, in Fig. 3h they become 10⁻¹³ / 10⁻¹⁶ A, while in Fig. S3 they are around 10⁻¹⁶ / 10⁻⁷ A. These inconsistencies need to be clarified and justified.

Version 1:

Reviewer comments:

Reviewer #1

(Remarks to the Author)

After careful assessment, the authors have addressed all of the concerns I raised. I therefore recommend that the manuscript be accepted for publication in Nature Communications.

Reviewer #3

(Remarks to the Author)

The authors have addressed my concerns.

Response to reviewers:

We sincerely thank the reviewers for the valuable comments, which have significantly improved the manuscript. Please find our point-by-point response to the comments below.

Response to Reviewer #1:

In this work, a 4.5 nm HfO_x switching layer is confined between an HfS₂ layer and a 3.5 nm elemental oxygen reservoir (EOR) layer. The device achieves ultrahigh Set/Reset switching speeds of 8 ns and 15 ns, respectively, with a normalized switching energy of 51 aJ, enabling an ultralow leakage current of approximately 0.05 fA/μm², a large on/off ratio exceeding 10¹⁰, and long retention endurance up to 10⁴ seconds. Furthermore, when modulated by pulse amplitude, the memristor demonstrates a high recognition accuracy of 97.0% in a convolutional neural network. Overall, this manuscript is well-organized and innovative. Considering that the overall quality, I recommend publication after a minor revision. Here are some suggestions for improvements:

Answer: We sincerely thank the reviewer for thoroughly examining our manuscript and for their valuable comments. We appreciate the recognition of the device structure, switching performance, ultralow leakage characteristics, and its potential for neuromorphic applications. We have carefully considered all the constructive suggestions provided and have revised the manuscript accordingly. Detailed point-by-point responses to each comment are included below. We believe that these revisions have further improved the clarity and quality of the manuscript.

1. The author can provide the electrical performance of the device when it is not oxidized? In Fig. 2, the influence of ozone treatment time on the electrical performance of the fabricated EOR-based memristors is provided. What would the performance of devices without oxidation be like?

Answer: We thank the reviewer for this valuable suggestion. We fabricated control devices without ozone treatment, as shown in Fig. R1a. The corresponding I-V curves (Fig. R1b) exhibit negligible resistive switching behavior. In our device design, the switching HfO_x layer is sandwiched between two 2D material layers (HfS₂ and MoS₂). Instead, the formation of conductive filaments driven by V_O is the fundamental operating mechanism of EOR memristors. However, the control MoS₂/HfS₂ stack was fabricated via mechanical exfoliation in a nitrogen-filled glove box, resulting in a very low density of vacancies and trapped charges. Without the ozone treatment to generate sufficient V_O, the device lacks the necessary conditions for filament formation, and the absence of the HfO_x tunneling barrier leads to low resistance. Consequently, no effective memristive window is observed. We have included these results and the corresponding discussion in the revised manuscript and Supplementary Information.

Fig. R1 | Electrical characteristics of the control MoS₂/HfS₂ device without ozone treatment. (a) Optical microscopy image. (b) I-V curves showing negligible resistive switching behavior.

2. What role does the following MoS₂ material play? To prevent the oxidation of MoS₂, the author deliberately selected a larger HfS₂ material. Why is HfS₂ placed on top of MoS₂ before oxidation? Oxidizing HfS₂ first and then performing van der Waals assembly won't result in a flatter interface?

Answer: We thank the reviewer for this valuable comment. First, the atomically flat surface of MoS₂ serves as an ideal template for the growth of the oxide switching layer. This minimizes reliability issues arising from thickness fluctuations, particularly in ultrathin films. Furthermore, this high-quality interface reduces trap states and suppresses unwanted charge trapping, which stabilizes the local electric field distribution and improves the uniformity of the switching process. The EOR layer originates from the vdW interface between HfS₂ and MoS₂, where the oxygen-rich environment promotes oxygen accumulation. Crucially, oxygen migration across the MoS₂ layer is energetically forbidden due to a large energy barrier (> 6 eV), and a high energy barrier of 1.21 eV for oxygen atoms to traverse from the HfO_x surface to the MoS₂ interface. Consequently, oxygen is confined between the HfO_x and MoS₂, which effectively suppresses leakage paths in the high-resistance state (HRS).

Regarding the fabrication process, the HfS₂ flake is designed to fully overlap the MoS₂ channel and contact regions. This avoids potential degradation from the ozone treatment and ensures the confinement of the EOR layer. In contrast, if the HfS₂ were oxidized prior to transfer, the resulting HfO_x surface would be directly exposed during van der Waals assembly. This surface is highly hydrophilic and prone to contamination by adsorbates. Additionally, since the oxidation of HfS₂ is not a self-limiting process, it leads to significant thickness variations in the HfO_x. This roughness is detrimental to controlling the uniformity of sub-10 nm films.

3. As shown in Fig. 2, the performance of the device is good. How about the reproducibility? What is the difference between the EOR layer and the hafnium oxide layer? How can the EOR layer affect the performance of the fabricated memristor?

Answer: We thank the reviewer for these constructive comments concerning device reproducibility and the role of the EOR layer. To systematically evaluate reproducibility, we fabricated 100 devices across four 5×5 arrays. The statistical distributions of key performance indicators, including off-state current, on-off ratio, and switching voltages, are presented in Fig. R2. These results demonstrate excellent device-to-device uniformity. Mechanistically, the EOR layer confines the HfO_x switching layer within a uniform structure and ensures a homogeneous electric field distribution, which is essential for reliable operation. Moreover, the EOR layer forms a pristine interface with the ultrathin oxide layer. It functions as a tunneling barrier to suppress the stochastic migration of V_O as well as leakage current. Consequently, the integration of the EOR layer with the switching oxide enables highly stable memristor operation with ultralow power consumption. We have added additional data and a detailed discussion in the revised manuscript.

Fig. R2 | Statistical electrical performance of 100 EOR-based memristors. (a) Superimposed I-V curves. (b) Statistical distributions of currents in the HRS and LRS. (c) On/off ratio distributions. (d) SET and RESET voltage distributions.

4. Fig. 4 shows the comparison of the recognition accuracy of MNIST test images. Why are the starting points of the accuracy for the three comparisons different? How does the EOR-based memristor maintain desirable linearity?

Answer: We thank the reviewer for this valuable comment. Different devices exhibit varying degrees of initial conductance uniformity. In ideal memristors, the standard deviation of initial conductance is negligible, allowing initialized weights to closely match the target values, thus yielding higher initial accuracy. In contrast, conventional memristors often show larger variations due to the stochastic distribution of oxygen

vacancies. This causes initialized weights to deviate from the ideal values, consequently degrading the initial accuracy. In our proposed device, both the thickness of the HfO_x switching layer and the oxygen concentration in the EOR layer are highly uniform. The EOR-based memristor exhibits stable analog switching characteristics. As a result, the conductance modulation is highly linear with respect to the applied programming current. Moreover, during repeated weight updates, the EOR layer acts as a reservoir to continuously replenish the oxygen atoms consumed in the HfO_x layer. Therefore, the abundant oxygen supply in the EOR layer facilitates precise control over the evolution of conductive filaments. This enhances the synaptic linearity, thereby increasing the recognition accuracy of the device.

5. It is not clear that how the ozone treatment is not affecting to MoS_2 ? How the authors control the treatment condition considering each sample has different HfS_2 thickness?

Answer: We thank the reviewer for this valuable comment. Although ozone treatment involves reactive species that could modify the surface, we designed the structure to prevent damage. Specifically, the HfS_2 flake fully overlaps the MoS_2 channel and contact regions to suppress inadvertently introduced defects and shield the MoS_2 from ultraviolet (UV) illumination, as indicated in Fig. R3a. Fig. R3b shows the Raman spectra from different regions of the device. The attenuation of the HfS_2 E_g/A_{1g} phonon modes confirms the complete transition to amorphous HfO_x . Additionally, the slight shift in the MoS_2 E^1_{2g}/A_{1g} peaks is attributed to the doping effect induced by the adjacent EOR layer. Our calculations indicate a high energy barrier of 1.21 eV for oxygen atoms to traverse from the HfO_x surface to the MoS_2 interface. Consequently, the accumulated oxygen tends to adsorb at the HfO_x interface, as shown in Fig. R3c. In our experiments, HfS_2 flakes with a thickness of 8-12 layers were selected to generate the HfO_x layer. By precisely controlling the ozone treatment duration, we can obtain an ultrathin oxide layer while preserving a residual thickness of HfS_2 . Since the device is a vertically stacked structure and the residual HfS_2 retains its single-crystal nature with excellent conductivity, slight thickness fluctuations have a negligible impact on the vertical current transport. Crucially, the HRS current and switching behavior are dominated by the thickness of the HfO_x switching layer, which is effectively tuned by the ozone treatment time.

Fig. R3 | (a) Optical microscopy image of a 5×5 memristor array. (b) Raman spectra from different

regions of the device. (c) Oxygen migration energy barrier calculated using the nudged elastic band (NEB) method.

6. In particular, there is evidence for a clean vdW interface and EOR layer, which are novel insight relevant to a broad audience and to the best of my knowledge the first demonstration of the usage EOR layer to improve the memristor performance, details about the fabrication processes and the TEM measurements should be added in method section.

Answer: We thank the reviewer for this valuable comment. The evidence for the EOR layer and the vdW interface is indeed a key strength of this work. Regarding the preparation of TEM samples: Briefly, an HfS₂ flake was placed onto the MoS₂ via a physical transfer method. Ozone treatment was then introduced to oxidize the HfS₂ into an HfO_x/HfS₂/HfO_x stack. Finally, a 50 nm Au film was deposited via thermal evaporation. The TEM sample preparation was performed using a JIB-4501 focused ion beam system, utilizing a liquid gallium (Ga) ion source. The procedure was as follows: First, the sample was securely mounted, and the region of interest (ROI) was located and aligned using the SEM mode. Coarse milling was then performed with a relatively high ion beam current to remove excess material and shape the lamella. Subsequently, the beam current was gradually reduced for fine milling. To achieve a high-quality surface finish suitable for atomic-resolution analysis, a final polishing step was conducted using a low-current ion beam. This step minimized surface damage and improved the clarity of the cross-section. We have updated the Methods section with these experimental details. Additionally, we have included detailed parameters for the TEM sample preparation and EELS/EDS mapping in the revised manuscript.

Fig. R4 | Cross-sectional cross profile of the TEM sample.

7. What are the parameters of the ozone treatment (i.e. chamber pressure, O₂ partial pressure, UV light intensity, ...)? How can it be ensured that the topmost MoS₂ layers are not oxidized, forming an MoO₃ interlayer? Free monoatomic oxygen is a radical which readily bonds to any nearby molecule. Thus, there are certainly no oxygen atoms at the HfO₂ surface, these are either O₂ molecules or ozone (O₃) molecules which are also highly reactive and thus likely bond to the HfO₂ surface. Thus, the enhanced concentration corresponds to molecules of either O₂, O₃ or an off-stoichiometric phase of hafnium oxide with an increased O concentration at the amorphous HfO₂ surface. Please modify the text in the main manuscript and the supplementary information

accordingly.

Answer: Ozone was generated via ultraviolet irradiation under ambient conditions using a 300 W UV lamp. The UV intensity was maintained at 50 mW/cm², resulting in a mild substrate temperature of 310 K. To prevent potential degradation of the underlying MoS₂ during this process, the HfS₂ flake was designed to fully overlap the MoS₂ channel and the contact regions.

We conducted a detailed depth-resolved XPS characterization of the EOR-based memristors to analyze the bonding states of oxygen. As illustrated in Fig. R5a, considering the XPS sampling depth of approximately 5 nm, the sulfur signal shows a significant increase after 2 min of etching, suggesting that the signal obtained at roughly 3 min corresponds to the HfO_x switching layer. When the etching time reaches 5 min, the Mo signal rises sharply, suggesting that the XPS spectra obtained at 6 minutes primarily originate from the EOR layer. Our analysis of the oxygen bonding states reveals that on the device surface (Fig. R5b), the content of high binding energy oxygen is relatively high, likely due to surface-adsorbed oxygen. In the HfO_x switching layer (Fig. R5c), the concentration of V_O increases while the high binding energy oxygen component decreases. Notably, the signal corresponding to the EOR layer is dominated by high binding energy components, while signals associated with V_O and Hf-O bonds diminish significantly. Combined with the EELS results, we attribute this layer primarily to O-O bonds (Fig. R5d), indicating the presence of molecular oxygen (O₂). Thermodynamically, the bond orders of peroxide (O₂²⁻, bond order = 1) and superoxide (O₂⁻, bond order = 1.5) are lower than that of neutral oxygen (O₂, bond order = 2). Consequently, ionic oxygen species are metastable and tend to react to achieve the stable O₂ electronic configuration; thus, we propose that the EOR layer is composed primarily of electroneutral oxygen. Additionally, due to the high V_O concentration in the HfO_x switching layer adjacent to the HfS₂ flake, a metallic hafnium phase is observed (Fig. R5e), which likely facilitates the forming-free operation of the device. Furthermore, as shown in Fig. R5f, after 6 min of etching, negligible Mo-O bond-related species are observed, confirming that no MoO₃ interlayer is formed. We have included this detailed analysis in the revised manuscript.

Fig. R5 | Depth-resolved XPS characterization. (a) Atomic concentration depth profiles as a function of etching time. (b–d) Depth-resolved O 1s XPS spectra obtained at (b) the surface, (c) 3 min, and (d) 6 min of etching. (e) Hf 4f spectrum after 3 min. (f) Mo 3d spectrum after 6 min.

Response to Reviewer #2:

The authors report a van der Waals (vdW) heterostructure memristor that integrates an ultrathin HfO_x switching layer and an “electroneutral elemental oxygen reservoir (EOR)” between MoS₂ and HfS₂. The device exhibits forming-free operation, an ultralow leakage current density of approximately 0.05 fA μm⁻², an on/off ratio exceeding 10¹⁰, and nanosecond-level switching speed (8 ns/15 ns) with a remarkably low switching energy of about 51 aJ. Such performance metrics are undoubtedly impressive and place the device among the best in the ultrathin or 2D-material-based memristor category. The manuscript is well-written, the experimental data are clean, and the proposed architecture is conceptually interesting. However, several central claims—particularly regarding the physical nature of the EOR layer and the mechanism of forming-free switching—are speculative and insufficiently supported by quantitative evidence. These gaps prevent the manuscript from reaching the level of mechanistic rigor expected in Nature Communications.

Answer: We would like to express our sincere gratitude to the reviewer for their thorough examination of our manuscript and for their valuable insights. We are particularly encouraged by the recognition of the device performance demonstrated in this study. Following the reviewer's suggestions, we have carefully addressed all comments to better highlight the innovation and novelty of our work, and have revised the manuscript accordingly.

In response to the concern regarding the physical nature of the EOR layer, we conducted additional depth-resolved XPS measurements and ELNES analysis to clarify the chemical bonding state of oxygen and substantiate our claims. A high energy barrier of 1.21 eV is required for an oxygen atom to traverse from the HfO_x surface to the MoS₂ interface, as calculated by the nudged elastic band (NEB) method. Consequently, the accumulated oxygen tends to adsorb onto the HfO_x switching layer. Moreover, the XPS signals from the EOR layer primarily exhibit peaks at high binding energies, while the signals for V_O and Hf-O bonds are significantly reduced. Considering the EELS results, we attribute this layer primarily to O-O bonds. This supports our claim that the EOR layer serves as an oxygen reservoir, thereby facilitating the rupture of conductive filaments in the high-resistance state (HRS).

In response to the concern regarding the forming-free switching mechanism, this behavior is attributed to the dynamic equilibrium of V_O facilitated by the EOR layer. By fitting the data to the Arrhenius equation, we estimated the activation energy for V_O migration to be 39.2 meV. Since the accumulated oxygen tends to adsorb onto the HfO_x switching layer, the V_O near the EOR layer undergo recombination, potentially rendering the effective thickness of the HfO_x switching layer thinner than its physical thickness (~4.5 nm). Furthermore, XPS results indicate abundant V_O and metallic hafnium within the HfO_x layer, leading to the partial formation of conductive paths. This pre-existing state reduces the operating voltage and facilitates forming-free behavior.

1. The main strength lies in its excellent device performance. The authors convincingly demonstrate that the vdW-derived EOR configuration can achieve extremely low leakage and high on/off ratio simultaneously, which has been a long-standing challenge in sub-5 nm oxide devices. The forming-free, ultrafast switching behavior and the low switching energy further highlight the potential of the approach for future low-power memory and neuromorphic computing. The device fabrication based on mild ozone oxidation of HfS₂ into HfO_x is elegant and avoids plasma damage or interface disorder typical of ALD or sputtered films. TEM and EELS analyses provide a visually persuasive structural picture of the vertical stack, and temperature-dependent I–V measurements help delineate conduction regimes. Overall, the data presentation is of high quality and suggests that the authors have achieved careful experimental control.

Answer: We sincerely thank the reviewer for the thorough assessment of our work and for recognizing the strengths of our device performance, data quality, and fabrication strategy. The detailed comments provided instrumental guidance for refining the manuscript and deepening the discussion of the underlying mechanisms. In response, we have carefully addressed these concerns through additional analyses and revised the manuscript accordingly.

2. Despite these, reviewer suggests that the mechanistic interpretation requires much stronger evidence. The manuscript repeatedly refers to the presence of an “electroneutral elemental oxygen reservoir” at the MoS₂/HfO_x interface, which is said to act as a tunneling barrier and diffusion regulator. However, the experimental basis for calling this layer “elemental” or “electroneutral” is insufficient. The EELS O–K edge data only reveal the presence of oxygen at the interface but do not identify its bonding state, valence, or neutrality. Without complementary depth-resolved XPS or ELNES analysis, it is impossible to confirm whether the layer consists of elemental oxygen, an oxygen-rich HfO_y, or MoO_y phase. This uncertainty is critical because the novelty and physical mechanism of the work hinge on the assumption that this layer is chemically distinct from conventional oxide reservoirs. At present, this interpretation remains speculative rather than conclusive.

Answer: We thank the reviewer for this important and constructive comment. In response to this concern, we have conducted depth-resolved XPS measurements and ELNES analysis to clarify the bonding states of oxygen in the memristor stack. As shown in Fig. R1a, given that the XPS sampling depth is approximately 5 nm, a significant increase in the S signal is observed after 2 min of etching. Therefore, the signal obtained at approximately 2 min corresponds to the HfS₂ layer. At an etching time of 3 min, the hafnium content begins to decrease while the oxygen concentration increases, identifying the HfO_x switching layer. After approximately 6 min of etching, the Mo content increases significantly, suggesting that the measurement region has probed into the EOR layer.

We analyzed the oxygen bonding states across different layers. On the device surface (Fig. R1b), the proportion of oxygen species at high binding energies is

relatively high (42%), which is likely ascribed to surface-adsorbed oxygen. In the HfO_x switching layer (Fig. R1c), the V_{O} concentration rises to 31%, while the high-binding-energy oxygen component (assigned to O-O bonds) decreases to 26%. Notably, the signal originating from the EOR layer (Fig. R1d) is predominately composed of the high-binding-energy component (reaching 52%), while the signals for V_{O} and Hf-O bonds decrease significantly. Therefore, in conjunction with the EELS results, we attribute this layer primarily to O-O bonds. In addition, bond order correlates with bond stability. A higher bond order implies greater bond energy and stability. The bond order of peroxide (O_2^{2-}) is 1, and that of superoxide (O_2^-) is 1.5, both of which are lower than that of neutral oxygen (O_2 , bond order = 2). Lower bond orders lead to reduced bond energies and thermodynamic instability. Consequently, O_2^{2-} and O_2^- species are metastable and highly reactive, tending to gain or lose electrons to achieve the more stable O_2 electronic configuration. Thus, based on the evidence of O-O bonding, we propose that the EOR layer is composed primarily of neutral oxygen. Furthermore, given the relatively high concentration of V_{O} in the HfO_x switching layer adjacent to the HfS_2 , a metallic hafnium phase is observed (Fig. R1e), which may facilitate the forming-free operation of the EOR-based memristor. As shown in Fig. R1f, negligible signals related to Mo-O bonds are detected after 6 min of etching, indicating that no MoO_3 interlayer is formed.

Fig. R1 | Depth-resolved XPS spectra. (a) Elemental atomic concentration profiles versus etch time. Depth-resolved O $1s$ XPS spectra measured (b) at the surface, and after (c) 3 min and (d) 6 min of etching. (e) Hf $4f$ XPS spectrum obtained after 3 min of etching. (f) Mo $3d$ XPS spectrum acquired after 6 min of etching.

The oxygen migration energy barrier was also calculated using the nudged elastic band (NEB) method, as shown in Fig. R2. A high energy barrier of 1.21 eV is required

for an oxygen atom to traverse from the HfO_x surface to the MoS_2 interface. Consequently, the accumulated oxygen tends to adsorb onto the HfO_x surface. This also explains the formation of the oxygen-rich HfO_x layer and the absence of Mo-oxide species. The O K-edge ELNES spectra acquired from the HfO_x switching layer and the EOR layer are shown in Fig. R2b. The O K-edge of the HfO_x is dominated by two peaks, labeled p_1 and p_2 , centered at 534.5 eV and 538.1 eV, respectively. The splitting between these two peaks is approximately 3.6 eV. Since the peak structure is sensitive to the presence of oxygen-related defects, the peaks appear broadened and less well-resolved [1]. As previously reported for transition metal oxide films with excess oxygen, the p_1 – p_2 peak splitting is typically absent; instead, the O K-edge of the EOR layer exhibits a distinct single peak at 538.3 eV [2–4]. Therefore, these results further confirm the presence of an oxygen-rich HfO_x layer and the EOR layer within the stack. We have added the corresponding description to the revised manuscript.

Fig. R2 | (a) Potential energy profile for oxygen migration calculated using the nudged elastic band (NEB) method. (b) O K-edge ELNES spectra acquired from the HfO_x switching layer and the EOR layer.

3. Similarly, the proposed “dynamic equilibrium between oxygen and vacancies” as the origin of forming-free switching is presented as a central mechanistic claim but is not directly proven. The data show forming-free I–V characteristics and repeatability, but no kinetic or temperature-dependent experiments are performed to extract activation energies or diffusion coefficients that would substantiate such an equilibrium model. Without such analysis, the claim of a self-equilibrating oxygen-vacancy process remains qualitative and speculative. Quantitative studies of pulse-width, temperature, or interval dependence are needed to establish whether the switching indeed arises from a balanced oxygen exchange rather than conventional filamentary behavior limited by thickness or compliance.

Answer: We thank the reviewer for this important and constructive comment. Temperature-dependent electrical measurements were conducted over the range of 77 K to 300 K to investigate the thermally activated transport of V_O . The activation energy was determined by fitting the current data to the Arrhenius equation: $I = I_0 \times \exp(-E_A/kT)$, where I is the current, I_0 is the pre-exponential factor, k is the Boltzmann constant, and T is the absolute temperature. The extracted activation energy was 39.2 meV.

Fig. R3 | (a) Arrhenius plot of $\ln I$ versus T^{-1} . (b) Dependence of the extracted slope Q on the bias voltage. Extrapolation to zero bias yields Q_0 , from which an activation energy of 39.2 meV is calculated.

To investigate the kinetics of forming-free switching, pulse-width-dependent measurements were performed by varying the pulse duration from 10 ns to 100 ns, as shown in Fig. R4. The results demonstrate that the current increases with increasing pulse width.

Fig. R4 | Read current of the EOR-based memristor measured at different pulse widths.

In our EOR-based memristor, the migration of V_O under an electric field drives the formation of conductive filaments (CFs). The EOR layer is strategically positioned between the 4.5 nm HfO_x switching layer and the MoS_2 , creating a nanoscale confined environment. This architecture effectively inhibits the stochastic diffusion of V_O , rendering the formation and rupture of CFs more deterministic, thereby facilitating stable forming-free switching. Moreover, the EOR layer at the HfO_x / MoS_2 interface leads to a locally modulated V_O concentration. This region serves as a functional oxygen reservoir that promotes the recombination of V_O with oxygen species during the reset process, resulting in an enhanced ON/OFF ratio and robust endurance. This dynamic equilibrium model aligns with mechanisms reported in prior studies, where forming-free characteristics are achieved by engineering an oxygen-rich interface to suppress the HRS current. Therefore, we adopt this model to explain the constrained filament growth, which accounts for the low-power LRS observed in our EOR-based memristors.

4. Another point of concern is the assertion that the EOR suppresses crosstalk in array configurations. The manuscript presents no actual crossbar array data, sneak-path analysis, or half-select stress experiments. The claim is extrapolated from single-device

leakage characteristics, which is not sufficient to demonstrate system-level selectivity or array compatibility. Similarly, while the manuscript attributes the ultralow leakage to the tunneling-barrier function of the EOR, temperature-dependent conduction analysis is limited to qualitative SCLC and Ohmic fitting. Quantitative extraction of barrier heights or Schottky/Poole–Frenkel/Fowler–Nordheim parameters would provide much stronger mechanistic grounding. Without these analyses, alternative explanations such as reduced trap density, series resistance in HfS_2 , or interface depletion cannot be excluded.

Answer: We appreciate the valuable suggestions. Accordingly, we conducted a systematic investigation of the memristive behavior in 5×5 crossbar arrays (Fig. R5). Representative bipolar I-V characteristics were obtained from 100 EOR-based memristors integrated into four separate 5×5 arrays. The highly consistent electrical performance indicates that the integrated HfO_x/EOR heterostructure effectively suppresses sneak-path currents, thereby mitigating crosstalk within the crossbar architecture. As suggested by the reviewer, a sneak-path analysis is provided in Fig. R5b. In conventional arrays without adequate leakage control, the programming of a selected cell can inadvertently disturb half-selected and unselected cells. The adopted voltage pulsing scheme involves applying the programming voltage to the selected cell, while half-selected cells are biased at a fractional voltage. Consequently, only the selected cell undergoes conductance switching upon the application of pulses, while the unselected cells in the array remain unperturbed (Fig. R5c).

Fig. R5 | Sneak-path analysis of the memristor arrays. (a) Optical image of a 5×5 memristor array. (b) Schematic illustration showing the definitions of different device states within the array. V_{prog} denotes the programming voltage applied to the word lines (WLs) to induce resistive switching in the selected cell. (c) Conductance evolution of selected, half-selected, and unselected cells in the array during pulse programming ($\pm 1.5 \text{ V}$, 20 ns).

In addition, specific cells within the array were programmed into the high-resistance state (HRS) to form predefined patterns (Fig. R6). The read currents of all cells were measured at a read voltage of 0.1 V and mapped onto a conductance heatmap. In this map, the programmed HRS cells are manifested as dark regions, corresponding to significantly reduced conductance, while the unprogrammed cells remain in the low-resistance state (LRS), appearing as light regions with high conductance. The distinct contrast between programmed and unprogrammed cells, along with the accurate

reproduction of the target patterns, demonstrates that the readout process is not compromised by sneak currents. This result confirms that crosstalk within the proposed memristor array is effectively suppressed, allowing each cell to be individually addressed and accessed without interference from adjacent devices.

Fig. R6 | Demonstration of digital pattern storage in a 5×5 crossbar memory array. The array exhibits programmed resistance states (gray for HRS and white for LRS). The designed pattern was written into the array using pulse programming. The x and y axes denote device coordinates (row and column indices), while the color scale (z-axis) represents the measured conductance values of the memory cells.

The extracted Schottky barrier height is 15.4 meV, as shown in Fig. R7. Consequently, this reduced barrier height lowers the energy threshold required for the formation of conductive filaments. Under an electric field, V_O are more susceptible to migration within the HfO_x switching layer. Furthermore, the presence of metallic Hf was detected in the XPS spectra of the HfO_x layer adjacent to the HfS_2 flake. This metallic component is a critical factor in realizing such a low interfacial barrier. Collectively, these factors enable the EOR-based memristor to exhibit forming-free

switching with minimal power consumption.

Fig. R7 | Schottky barrier height extraction for the EOR-based memristor. (a) Arrhenius plot showing the dependence of IT^{-2} on inverse temperature T^{-1} , where the solid lines represent linear fits. (b) Dependence of the extracted slope S on the bias voltage. Extrapolation to zero bias yields S_0 , which is utilized to calculate the effective Schottky barrier height.

5. The reliability data are also limited relative to the claims. The authors describe “robust retention and endurance,” yet only 10^3 cycles and retention up to 10^5 s (~ 28 h) at room temperature are presented. These values, though reasonable for a proof-of-concept, do not justify the use of “robust” or “stable” at the level expected for nonvolatile memory publications in Nature Communications. Measurements of endurance up to 10^5 – 10^6 cycles, retention at elevated temperatures (85–125 °C), and read-disturb or bias-stress testing are necessary to validate the long-term stability claims.

Answer: We appreciate the valuable suggestions. We agree that retention at elevated temperatures and extended endurance testing are crucial for a rigorous assessment of nonvolatile memory reliability. While we initially reported endurance data up to 10^3 cycles as a proof-of-concept, we have now extended the testing to 10^5 cycles, as shown in Fig. R8a. These additional measurements demonstrate that the device exhibits excellent endurance without significant performance degradation. A stable ON/OFF ratio and small fluctuation in switching characteristics was observed throughout the cycling process. These results strongly support our claim of “robust endurance” and confirm that the device is capable of prolonged operation, a key requirement for nonvolatile memory applications. We have also addressed the concern regarding data retention by conducting tests at elevated temperatures. Specifically, retention characteristics were measured at 85 °C and 125 °C, as shown in Fig. R8b and R8c. The results demonstrate that the device retains its memory states for over 10^5 s, even at these elevated temperatures. This retention performance is consistent with the behavior observed at room temperature and validates the device’s thermal stability—a critical requirement for practical applications in harsh environments.

Fig. R8 | Endurance and retention performance of the EOR-based memristor. (a) Endurance characteristics monitored over 10^5 cycles using bipolar voltage pulses of ± 1.5 V (pulse width: 200 μ s). (b, c) Retention characteristics measured at 85 °C and 125 °C, respectively.

6. The comparison with the ALD-HfO_x control device is helpful but not conclusive, as the two structures differ in fabrication method, oxidation time, and possibly thickness or interface roughness. Therefore, attributing the performance improvement solely to the EOR effect is premature. A head-to-head comparison with strictly identical geometries and electrode materials would be required to isolate the role of the EOR. Furthermore, the neuromorphic functionality—highlighted by a 97% MNIST recognition accuracy—is purely simulation-based. While the LTP/LTD data are experimentally measured, the neural network performance results from an offline simulation that does not include device-to-device variation, cycle-to-cycle drift, or retention degradation. Without hardware-in-the-loop or on-chip learning experiments, the neuromorphic claims should be presented more cautiously as “potential applications” rather than demonstrations.

Answer: We appreciate the valuable suggestions. We agree with the reviewer that the ALD- HfO_x control device differs significantly from our EOR-based memristor in terms of fabrication methodology, oxidation duration, thickness, and interface roughness. Consequently, a direct comparison with the ALD-based device may lack sufficient rigor within the context of this study. Accordingly, we have removed these data and the corresponding descriptions from the revised manuscript.

Figure 9 illustrates the thickness evolution of HfS₂ during its conversion to HfO_x via ozone oxidation. With increasing ozone treatment time, the HfS₂ flake gradually transforms from a partially oxidized state to a completely converted HfO_x layer. Based on the initial thickness of HfS₂ (11.17 nm, Fig. R9b) and the final thickness of the fully oxidized sample (15.13 nm, Fig. R9i) measured by atomic force microscopy, the expansion ratio was determined to be 1.35. To quantify the thickness of the converted HfO_x at various stages, we defined two parameters: the total AFM-measured thickness (t) and the thickness of the residual unconverted HfS₂ (x). Their relationship is expressed by the following equation: $t = x + 1.35 \times (11.17 - x)$. By determining the value of x , the thickness of the formed HfO_x layer can be precisely derived. This indicates that increasing the ozone treatment time can increase the thickness of the HfO_x layer.

Fig. R9 | Controllable oxidation of the HfO_x. (a) Total thickness of the HfO_x/HfS₂/HfO_x stack and the HfO_x thickness as a function of ozone treatment time. (b-i) Evolution of height profiles with varying ozone treatment time. Each inset shows an AFM image. Scale bar: 2 μm.

To elucidate the influence of switching layer thickness and the EOR layer on device performance, a direct, systematic comparison using devices with identical geometries and electrode materials is provided in Fig. R10. Since the oxidation of HfS₂ into HfO_x is not a self-limiting process, prolonging the oxidation time (ozone treatment duration) results in an increased HfO_x thickness. As the oxidation duration exceeds 40 min, the high-resistance state (HRS) current exhibits a continuous rise. This is primarily attributed to the enhanced presence of remnant conductive filaments within the thicker switching layer, which serve as leakage paths and lead to a higher HRS current. Moreover, the magnitudes of V_{SET} and V_{RESET} increase, as a thicker switching layer requires a higher electric field to induce resistive switching. Therefore, reducing the thickness of the switching layer is effective in lowering the operating voltages. Simultaneously, the introduction of the EOR layer facilitates the attainment of an extremely low HRS current.

Fig. R10 | Impact of HfO_x switching layer thickness on device performance. (a) Influence of ozone treatment duration on the electrical characteristics of the EOR-based memristor. (b) Dependence of LRS and HRS currents on oxidation duration. (c) Dependence of switching voltages (V_{SET} and V_{RESET}) on oxidation duration.

7. In comparison to recent literature, the manuscript distinguishes itself by proposing a vdW-induced, neutral-oxygen interlayer rather than the conventional trilayer HfO_x/HfO₂/HfO_x or HfO₂/ZrO_{2-x}/HfO₂ stacks used as oxygen-vacancy reservoirs. Chemically, this approach could represent a new direction for interfacial control in 2D memristors. However, recent trilayer studies already demonstrate CMOS-compatible fabrication, array-level validation, and selector-free suppression of sneak paths—advantages that this work does not yet match. The novelty of a “neutral oxygen” reservoir is intriguing but must be proven chemically to stand as a clear advance over these prior works.

Answer: We appreciate the reviewer’s valuable suggestions. Regarding the spatial distribution of oxygen, the proposed EOR-based memristor aligns with the design principles of previously reported devices constructed by engineering oxygen concentration gradients across heterostructures, such as HfO_x/HfO₂/HfO_x or HfO₂/ZrO_{2-x}/HfO₂ stacks. To further substantiate this, we have included depth-resolved XPS and ELNES analyses to elucidate the chemical nature of oxygen within the EOR layer. The results demonstrate that the EOR layer functions as an effective oxygen reservoir.

8. In conclusion, this manuscript presents a technically strong and potentially impactful concept with outstanding device performance. However, several mechanistic and interpretive claims are currently speculative or insufficiently evidenced. The work would benefit greatly from deeper chemical analysis of the EOR layer, quantitative conduction modeling, comprehensive reliability testing, and hardware-level neuromorphic validation. If these issues are addressed, the study could reach the broad and rigorous standard of Nature Communications. At its current stage, I recommend major revision. The authors are encouraged to provide quantitative and chemical evidence to solidify their mechanistic interpretation and to moderate claims that are presently speculative.

Answer: We appreciate the reviewer's valuable suggestions. To address the concerns regarding the physical nature of the EOR layer, we have provided depth-resolved XPS

analysis to quantify the elemental distribution and chemical states within the EOR layer. Regarding the switching mechanism, the spatial oxygen distribution in our EOR-based memristor aligns with previously reported multi-layer devices. We attribute the superior device performance—specifically the low HRS current and ultra-low power consumption—to the synergistic effect between the ultrathin HfO_x switching layer and the electroneutral EOR layer. Following the reviewer's suggestion, a comprehensive sneak-path analysis has been conducted to demonstrate the reliable operation of the EOR-based crossbar array. Furthermore, we have clarified all ambiguous statements throughout the revised manuscript. We sincerely thank you for your constructive and invaluable feedback, which has helped us to substantially refine the quality of our work.

References

1. X. F. Wang, Quan Li, R. F. Egerton, P. F. Lee, J. Y. Dai, Z. F. Hou, X. G. Gong. Effect of Al addition on the microstructure and electronic structure of HfO₂ film. *J. Appl. Phys.* **1** 013514 (2007).
2. Shriram Ramanathan, David A. Muller, Glen D. Wilk, Chang Man Park, Paul C. McIntyre. Effect of oxygen stoichiometry on the electrical properties of zirconia gate dielectrics. *Appl. Phys. Lett.* **20** 3311–3313 (2001).
3. Busch, B.W., Pluchery, O., Chabal, Y.J. *et al.* Materials Characterization of Alternative Gate Dielectrics. *MRS Bull.* **27** 206–211 (2002).
4. Stemmer, S., Chen, Z.Q., Zhu, W.J. and Ma, T.P. Electron energy-loss spectroscopy study of thin film hafnium aluminates for novel gate dielectrics. *J. Microsc.* **210** 74–79 (2003).

Response to Reviewer #3:

This manuscript investigates HfS₂-based memristive devices. However, the work does not demonstrate substantial advances in either device performance or mechanistic understanding. Although the paper claims a sub-5-nm switching region, this refers merely to the film thickness rather than the actual device footprint. I initially expected lateral device scaling, but the reported vertical dimension does not provide meaningful novelty. The manuscript further claims “high-speed” switching, including the 8 ns / 15 ns switching times emphasized in the title and abstract. In reality, the devices are driven by 100 ns pulses, with the current merely reaching a steady level at 8 ns / 15 ns. This does not constitute stable switching. In the community, a switching speed is typically accepted only if the device can reliably toggle under pulses of that duration, and the subsequent voltage/current holding sequence confirms stable filament formation or rupture. The same issue applies to the purported “ultralow energy” operation, where only the portion of the current transient up to stabilization is considered—this is not an appropriate metric. The MNIST recognition demonstration based on simulations also does not introduce meaningful novelty.

Answer: We sincerely thank the reviewer for their thorough examination of our manuscript and for their valuable comments. Detailed point-by-point responses are provided below. We believe that these revisions have significantly improved the clarity and quality of the manuscript. Additionally, we have revised the Introduction section to better highlight the novelty of this work. We have also addressed specific points regarding the advancements of our work, particularly in terms of structural design, device performance, and mechanistic insights.

Regarding the advantages of thickness scaling, the reviewer correctly notes that the "sub-5 nm switching region" refers to the vertical thickness of the HfO_x functional layer rather than the lateral device footprint. While lateral scaling is a standard approach in CMOS technology to increase integration density, vertical scaling of the functional layer offers a viable alternative route to minimize the operating voltage for low-power applications. In contrast, aggressive lateral scaling often suffers from edge effects, increased leakage paths, and non-uniform electric fields, all of which can degrade switching reliability. However, thickness scaling presents significant challenges. As the thickness of the switching layer decreases, issues related to the stochastic growth and rupture of conductive filaments (CFs) and interface stability become increasingly prominent. A primary concern is that thickness variations and surface roughness significantly impact the electric field distribution across the device—particularly when the thickness falls below 10 nm—thereby increasing the variability of switching parameters. In addition, the difficulty in precisely controlling the dissolution of CFs often leads to the presence of remnant filaments, which serve as leakage paths and result in a high high-resistance state (HRS) current and increased static power consumption. Moreover, ultra-thin switching layers are highly susceptible to interdiffusion with the electrodes, a phenomenon that severely degrades device stability, reliability, and cycling endurance.

Advantages of the proposed memristor structure: By exploiting the atomically flat interfaces of 2D materials, a uniform, ultrathin HfO_x switching layer is formed between the HfS₂ and MoS₂ layers. Consequently, the migration of V_O is strictly confined within a nanoscale region (sub-10 nm). This architecture ensures a homogeneous electric field distribution and facilitates the vertical scaling of the switching layer. Furthermore, the EOR layer and the underlying HfS₂ flake act as effective diffusion barriers that inhibit the stochastic diffusion of V_O, resulting in robust endurance and retention performance for the EOR-based memristors.

Novelty in the operational mechanism: The low high-resistance state (HRS) current originates from the spontaneously formed EOR layer, in distinct contrast to the intricate engineering of oxygen concentration gradients using multi-layer stacks seen in previous reports. The electroneutral EOR layer, adjacent to the ultrathin HfO_x switching layer, facilitates a dynamic equilibrium of V_O. This mechanism promotes the rupture of conductive filaments (CFs) and serves as an effective leakage barrier in the HRS. Consequently, the EOR-based memristors achieve ultrafast operation and an extremely low HRS current. Furthermore, the high V_O concentration and the presence of metallic hafnium within the ultrathin HfO_x switching layer enable forming-free operation.

Figure R1 illustrates the thickness evolution of HfS₂ during its conversion to HfO_x via ozone oxidation. With increasing ozone treatment time, the HfS₂ flake gradually transforms from a partially oxidized state to a completely converted HfO_x layer. Based on the initial thickness of HfS₂ (11.17 nm, Fig. R1b) and the final thickness of the fully oxidized sample (15.13 nm, Fig. R1i) measured by atomic force microscopy, the expansion ratio was determined to be 1.35. To quantify the thickness of the converted HfO_x at various stages, we defined two parameters: the total AFM-measured thickness (t) and the thickness of the residual unconverted HfS₂ (x). Their relationship is expressed by the following equation: $t=x+1.35\times(11.17-x)$. By determining the value of x , the thickness of the formed HfO_x layer can be precisely derived. This indicates that increasing the ozone treatment time can increase the thickness of the HfO_x layer.

Fig. R1 | Controllable oxidation of the HfO_x. (a) Total thickness of the HfO_x/HfS₂/HfO_x stack and the HfO_x thickness as a function of ozone treatment time. (b-i) Evolution of height profiles with varying ozone treatment time. Each inset shows an AFM image. Scale bar: 2 μm.

To elucidate the influence of switching layer and EOR layer thickness on memristor performance, a systematic comparison using devices with identical lateral geometries and electrode materials is presented in Fig. R2. Since the oxidation of HfS₂ into HfO_x is not a self-limiting process, prolonging the oxidation duration (ozone treatment time) leads to an increased HfO_x thickness. As the oxidation duration exceeds 40 min, the high-resistance state (HRS) current exhibits a continuous rise. This is primarily attributed to the increased presence of remnant conductive filaments within the thicker switching layer, which serve as leakage paths and lead to a higher HRS current. Moreover, the magnitudes of V_{SET} and V_{RESET} increase, as a thicker switching layer requires a higher electric field to induce resistive switching. Therefore, reducing the thickness of the switching layer is effective in lowering the operating voltage, thereby promoting low-power operation. Simultaneously, the introduction of the EOR layer enables the attainment of extremely low HRS currents.

Fig. R2 | Impact of HfO_x switching layer thickness on device performance. (a) Influence of ozone treatment duration on the electrical characteristics of the EOR-based memristor. (b) Dependence of LRS and HRS currents on oxidation duration. (c) Dependence of switching voltages (V_{SET} and V_{RESET}) on oxidation duration.

Dynamic switching operations were performed using 100 ns voltage pulses, with the current stabilizing within 8 ns and 15 ns during the Set and Reset processes, respectively. These ultra-short intervals represent the intrinsic Set/Reset transition times of the memristor [1, 2]. Calculating power consumption based solely on these transitions reflects the instantaneous transition power, rather than the total energy consumed during a complete pulse cycle (which includes the non-switching duration). However, for practical applications, such a metric lacks practical engineering relevance. Moreover, despite variations in device structure, materials, and dimensions across different technologies, low leakage current remains a critical and universally comparable performance indicator. This metric is particularly vital when evaluating the power budget of circuit-level applications. Therefore, to ensure rigorous reporting, we have removed the broad claim of "low power consumption" and instead prioritized the specific discussion of "low HRS current" in the revised manuscript.

The MNIST dataset serves as a cornerstone benchmark in computer vision for evaluating image classification algorithms. Simulations utilizing this dataset are commonly employed to assess the potential of emerging hardware for artificial intelligence applications. The primary purpose of our simulation was to highlight the performance advantages of the EOR-based memristor. Although a recognition accuracy of 97% was achieved, we agree with the reviewer that the simulation did not account for device non-idealities, such as device-to-device variation, cycle-to-cycle variability, or retention degradation. Consequently, we have tempered our claims regarding neuromorphic computing, presenting them as "potential applications" rather than fully realized demonstrations. The manuscript has been revised accordingly.

In summary, the vertical sub-5 nm switching layer represents a significant advancement in device miniaturization. Furthermore, the EOR layer offers new mechanistic insights into forming-free, low-leakage resistive switching behaviors.

1. Device area: The manuscript does not specify the device area, which is critical for evaluating performance. Only current density is reported, without the absolute current values.

Answer: We appreciate the valuable suggestions. The active area of each memristor is defined by the geometric overlap between the top and bottom electrodes, measuring $4.2 \mu\text{m} \times 4.2 \mu\text{m}$ (Fig. R3a). Consequently, the current density was calculated by normalizing the measured current against this effective area ($17.64 \mu\text{m}^2$). We fully agree

with the reviewer that providing both absolute current values and specific device dimensions is essential for a rigorous performance assessment. Therefore, we have explicitly stated these parameters in the revised manuscript. Furthermore, to ensure completeness, I-V characteristics plotting both the absolute current and the normalized current density have been included in the Supplementary Information.

Fig. R3 | Electrical performance of the EOR-based memristor array. (a) Optical image of the fabricated 5×5 crossbar memristor array. (b) Typical I-V curves showing the absolute current. (c) Corresponding I-V curves plotted as normalized current density.

2. Endurance testing: Endurance is typically evaluated using $\sim 200 \mu\text{s}$ pulses. What would the endurance look like under 8 ns/15 ns pulses? The reported endurance of $\sim 1,000$ cycles is insufficient for practical applications; can it be improved?

Answer: We appreciate the reviewer's insightful inquiry regarding the device's response to nanosecond-scale programming pulses. To address this, we investigated the conductance modulation induced by electrical pulses with durations ranging from 10 ns to 40 ns, as shown in Fig. R4. Upon the application of these nanosecond-scale pulses, the memristor exhibited clear and reproducible conductance states, confirming that the switching mechanism can be effectively activated on a much shorter timescale. To investigate the kinetics of forming-free switching, pulse-width-dependent measurements were performed by varying the pulse duration from 10 ns to 100 ns (Fig. R4e). The results demonstrate that the current increases with increasing pulse width. Building on these insights, we further evaluated the endurance performance using 100 ns programming pulses. The device demonstrated stable and repeatable switching for at least 10^5 cycles, with negligible degradation in current levels (Fig. R4f). These results confirm that while microsecond pulses induce a larger conductance modulation, the device remains fully functional under nanosecond-scale pulses, and reliable endurance is maintained even with 100 ns pulses. The corresponding experimental results and discussion have been incorporated into the revised manuscript to elucidate the pulse-width-dependent programming and endurance behavior.

Fig. R4 | Conductance response to programming pulses with varying durations. (a-d) Conductance response to 1.5 V pulses with durations of (a) 10 ns, (b) 20 ns, (c) 30 ns, and (d) 40 ns. (e) Conductance modulation induced by pulses with durations ranging from 10 ns to 100 ns. (f) Corresponding endurance characteristics measured using SET/RESET pulses of ± 1.5 V (100 ns duration).

3. Retention testing: Retention should not be evaluated only at room temperature. Measurements at 85 °C—or preferably 125 °C—are standard practice.

Answer: We appreciate the reviewer's valuable suggestion regarding retention characterization at elevated temperatures. To address this point, we conducted retention measurements at 85 °C and 125 °C, which are critical benchmarks for evaluating the long-term stability of non-volatile memory devices. The results demonstrate that the device exhibits excellent retention performance under thermal stress (Fig. R5). Both the HRS and LRS remained stable over a duration of 10^5 s, indicating that the switching states are thermally robust and exhibit negligible degradation at these elevated temperatures. We have incorporated these data and the corresponding discussion into the revised manuscript.

Fig. R5 | Retention characteristics of the EOR-based memristor at elevated temperatures. (a) Retention measured at 85 °C. (b) Retention measured at 125 °C.

4. Memory window: The experimental results indicate a memory window of approximately 3–5 orders of magnitude, with noticeable variation across tests. This is far from the 10-order-of-magnitude window claimed in the manuscript.

Answer: We appreciate the valuable suggestion. To ensure statistical rigor, we extracted the HRS and LRS currents from the I-V curves of 100 devices (measured at $V_{\text{read}} = 0.1$ V) and calculated their ON/OFF ratios. As shown in Fig. R6, while a maximum switching ratio of 10^{10} was observed, we opted for a more conservative evaluation. Based on the statistical analysis, the average HRS current density is $1.52 \text{ fA}/\mu\text{m}^2$, while the average LRS current density is $0.27 \mu\text{A}/\mu\text{m}^2$. Therefore, to more accurately reflect the reliability and reproducibility of the device performance, we report an average ON/OFF ratio of 1.47×10^9 . We have updated the manuscript to reflect these statistical performance metrics.

Fig. R6 | Statistical switching performance of the EOR-based memristors. (a) Cumulative probability distributions of the HRS and LRS currents. (b) Cumulative probability distribution of the on/off ratio extracted at $V_{\text{READ}} = 0.1 \text{ V}$.

5. Inconsistencies in resistance values: The reported high-/low-resistance states vary significantly across Figs. In Fig. 2b they are $10^{-12} / 10^{-16} \text{ A}$, in Fig. 3h they become $10^{-13} / 10^{-16} \text{ A}$, while in Fig. S3 they are around $10^{-16} / 10^{-7} \text{ A}$. These inconsistencies need to be clarified and justified.

Answer: We appreciate the reviewer's careful examination of the resistance values reported in the manuscript. The observed variations originated from differences in sample sizes across the datasets. Specifically, the original Fig. 2b, which depicts the evolution of current as a function of oxidation duration, presented average values derived from an initial subset of 25 devices within a single 5×5 array. This relatively small sample size accounted for the previously reported lower average HRS current. In contrast, Fig. R6a presents the statistical distribution of HRS and LRS currents collected from 100 devices. To ensure accuracy and statistical representativeness, we have recalculated the average values and error bars for the 40-minute ozone treatment group using this larger dataset. Fig. 2b has been updated in the revised manuscript to reflect these corrected statistics.

Regarding Fig. 3h, which presents temperature-dependent current characteristics, the data were acquired at a read voltage of $V_{\text{Read}}=0.2$ V, which differs from the condition used in Fig. 2b. We selected $V_{\text{Read}} = 0.2$ V for this measurement because the temperature-dependent variation in HRS current is less distinguishable at 0.1 V due to the low current level. A higher read voltage allows for a clearer visualization of the conduction mechanisms. To prevent ambiguity, we have explicitly specified the read voltage in both Fig. 3h and its corresponding caption in the revised manuscript. Furthermore, we acknowledge that the initial temperature-dependent data were obtained using an Agilent 2912 semiconductor analyzer, which has a higher noise floor compared to the Keithley 4200-SCS used for testing the afore mentioned I-V curves. The background noise current measured using the two instruments is shown in the Fig. R7. This was the primary reason for the discrepancy in off-state current values. However, the evolution trend of the electrical performance of the device with temperature is the same. To ensure total data consistency across the manuscript, we have re-measured the temperature-dependent performance using the Keithley 4200-SCS and updated Fig. 3h accordingly.

Figure R6a presents the statistical current distributions collected from 100 devices. The calculated average current densities for the HRS and LRS are 1.52 fA/ μm^2 and 0.27 $\mu\text{A}/\mu\text{m}$, respectively, which align closely with the values noted by the reviewer. Furthermore, these results are consistent with the other experimental data presented in the manuscript.

We sincerely thank you for your meticulous examination of our manuscript. We have revised the paper significantly to incorporate your constructive comments.

Fig. R7 | Background noise current of the (a) 4200A-SCS and (b) B2912A semiconductor analyzers, respectively.

References

1. Teja Nibhanupudi, S.S., Roy, A., Veksler, D. *et al.* Ultra-fast switching memristors based on two-dimensional materials. *Nat. Commun.* **15**, 2334 (2024).
2. Wang, G., Li, S., Wang, C. *et al.* Structural Transformation of 2D InSe Toward Ultrafast and Energy-Efficient Non-Volatile Memristive Switching. *Adv. Funct. Mater.* e16141 (2025).